# Genetically diverse uropathogenic *Escherichia coli* adopt a common transcriptional program in patients with UTIs

Anna Sintsova[1], Arwen E Frick-Cheng[1], Sara Smith[1], Ali Pirani[1], Sargurunathan Subashchandrabose[2], Evan S Snitkin[1], Harry Mobley[1]*

[1]Department of Microbiology and Immunology, University of Michigan, Ann Arbor, United States; [2]Department of Veterinary Pathobiology, Texas A&M University, College Station, United States

**Abstract** Uropathogenic *Escherichia coli* (UPEC) is the major causative agent of uncomplicated urinary tract infections (UTIs). A common virulence genotype of UPEC strains responsible for UTIs is yet to be defined, due to the large variation of virulence factors observed in UPEC strains. We hypothesized that studying UPEC functional responses in patients might reveal universal UPEC features that enable pathogenesis. Here we identify a transcriptional program shared by genetically diverse UPEC strains isolated from 14 patients during uncomplicated UTIs. Strikingly, this in vivo gene expression program is marked by upregulation of translational machinery, providing a mechanism for the rapid growth within the host. Our analysis indicates that switching to a more specialized catabolism and scavenging lifestyle in the host allows for the increased translational output. Our study identifies a common transcriptional program underlying UTIs and illuminates the molecular underpinnings that likely facilitate the fast growth rate of UPEC in infected patients.
DOI: https://doi.org/10.7554/eLife.49748.001

*For correspondence:
hmobley@med.umich.edu

Competing interests: The authors declare that no competing interests exist.

## Introduction

Urinary tract infections (UTIs) are among the most common bacterial infections in humans, affecting 150 million people each year worldwide (*Flores-Mireles et al., 2015*). A high incidence of recurrence and frequent progression to chronic condition exacerbates the negative impact of UTIs on patients' quality of life and healthcare cost (*Foxman, 2010*). Despite the magnitude of the problem, treatment remains limited by a strain's susceptibility to available antibiotics, which are often ineffectual (*Albert et al., 2004*; *Nickel, 2005*; *Sihra et al., 2018*).

The major causative agent of uncomplicated UTIs is Uropathogenic *Escherichia coli* (UPEC), which is responsible for upwards of 70% of all cases (*Flores-Mireles et al., 2015*). The majority of our insights into UPEC pathogenesis have been obtained through in vitro assays, cell culture systems, and animal models (*Alteri et al., 2009*; *Alteri and Mobley, 2015*; *Sivick and Mobley, 2010*; *Subashchandrabose and Mobley, 2015*). While these studies have identified virulence and fitness factors that are important for UPEC infection, how these studies translate to human infection is not clear. As a result, we do not yet have a complete understanding of UPEC physiology in the human urinary tract. Moreover, the genetic heterogeneity of UPEC isolates, which carry diverse and functionally redundant virulence systems including iron acquisition, adherence, and toxins, further complicates our understanding of uropathogenesis (*Johnson et al., 1998*; *Johnson et al., 2001*; *Köhler and Dobrindt, 2011*; *Schreiber et al., 2017*; *Takahashi et al., 2006*). The different

constellations of virulence factors and diverse genetic backgrounds raise the question of whether different UPEC strains vary in their strategies for pathogenesis.

Since defining conserved UPEC characteristics have proven elusive to comparative genomics strategies, we hypothesized that comparing functional responses in the context of the host may uncover disease-defining features. To that end, we examined UPEC gene expression directly from 14 patients with documented significant bacteriuria and presenting with uncomplicated UTI and compared it to the gene expression of the identical strains cultured to mid-exponential stage in filter-sterilized pooled human urine. Despite the genetic diversity of the pathogen and the human hosts, we identified a remarkably conserved gene expression program that is specific to human infection and strongly supports previous findings of extremely rapid UPEC growth rate during UTI (*Bielecki et al., 2014*; *Burnham et al., 2018*; *Forsyth et al., 2018*). Importantly, we show that this transcriptional program is recapitulated in the mouse model of infection and propose a mechanism by which the fast growth rate can be achieved. Based on extensive analysis, we propose a model where UPEC shut down all non-essential metabolic processes and commit all available resources to rapid growth during human UTI. Critically, our discovery of a common transcriptional program of UPEC in patients significantly expands our understanding of bacterial adaptation to the human host and provides a platform to design universal therapeutic strategies.

## Results

### Study design

To better understand UPEC functional responses to the human host, we isolated and sequenced RNA from the urine (stabilized immediately after collection) from fourteen otherwise healthy women diagnosed with UPEC-associated urinary tract infection. To identify infection-specific responses, we cultured the same fourteen UPEC isolates in vitro in filter-sterilized human urine (mid-exponential phase, 2 hr time point in *Figure 1—figure supplement 1*), and isolated and sequenced RNA from these cultures (study design and quality control is described in detail in Methods section). Phylogenetic analysis showed a high degree of genetic diversity, as we identified strains belonging to three distinct phylogroups, 13 different sequence types, and 13 distinct serogroups (*Figure 1—figure supplement 2*, *Table 1*, *Table 2*). The majority of UPEC isolates (10 of 14) belonged to the B2 phylogroup, which is consistent with previously published studies (*Foxman, 2010*; *Schreiber et al., 2017*). Although the majority (10 of 14) of patients had a previous history of UTIs, we found no relationship between patients' previous UTI history and bacterial genotype (*Figure 1—figure*

**Table 1.** Sequence type for 14 clinical UPEC isolates

| Strain | Sequence type | Adk | fumC | gyrB | Icd | Mdh | purA | recA |
|--------|--------------|-----|------|------|-----|-----|------|------|
| HM01 | 69 | 21 | 35 | 27 | 6 | 5 | 5 | 4 |
| HM03 | 101 | 43 | 41 | 15 | 18 | 11 | 7 | 6 |
| HM06 | 131 | 53 | 40 | 47 | 13 | 36 | 28 | 29 |
| HM07 | 641* | 9 | 6 | 33* | 131 | 24 | 8 | 7 |
| HM14 | Novel | 6 | 4 | 4 | 16 | 24 | 13 | 14 |
| HM17 | 73 | 36 | 24 | 9 | 13 | 17 | 11 | 25 |
| HM43 | Novel* | 40* | 14 | 19 | 36 | 17 | 10 | 203 |
| HM54 | 404* | 14* | 14 | 10 | 14 | 17 | 7 | 74 |
| HM56 | 538 | 13 | 40 | 19 | 13 | 36 | 28 | 30 |
| HM57 | 73 | 36 | 24 | 9 | 13 | 17 | 11 | 25 |
| HM60 | 648 | 92 | 4 | 87 | 96 | 70 | 58 | 2 |
| HM66 | 80 | 13 | 24 | 19 | 14 | 23 | 1 | 10 |
| HM68 | 998 | 13 | 52 | 156 | 14 | 17 | 25 | 17 |
| HM86 | 127 | 13 | 14 | 19 | 36 | 23 | 11 | 10 |

DOI: https://doi.org/10.7554/eLife.49748.002

**Table 2.** In silico determined serotypes for 14 clinical UPEC strains

| Strain | H_type | O_type |
| --- | --- | --- |
| HM01 | H4 | O25 |
| HM03 | H21 | NA |
| HM06 | H4 | O25 |
| HM07 | H45 | O45 |
| HM14 | H10 | O8 |
| HM17 | H1 | O6 |
| HM43 | H23 | NA |
| HM54 | H5 | O75 |
| HM56 | H4 | O13/O135 |
| HM57 | H1 | O2/O50 |
| HM60 | H10 | O102 |
| HM66 | H7 | O7 |
| HM68 | H6 | O2/O50 |
| HM86 | H31 | O6 |

DOI: https://doi.org/10.7554/eLife.49748.003

supplement 2). Moreover, the 14 clinical isolates showed a wide array of antibiotic resistance phenotypes (*Figure 1—figure supplement 2*).

## Virulence factor expression is observed both during urine culture and human infection

We first assessed the virulence genotype of the fourteen UPEC strains by looking at the presence or absence of a comprehensive list of known virulence factors, including adhesins, toxins, iron acquisition proteins, and flagella (*Johnson et al., 2001*; *Johnson and Stell, 2000*; *Köhler and Dobrindt, 2011*; *Schreiber et al., 2017*; *Subashchandrabose and Mobley, 2015*; *Tarchouna et al., 2013*) (*Figure 1A*). As previously reported (*Schreiber et al., 2017*), B1 strains appear to carry fewer virulence factors overall when compared to B2 strains, suggesting that UTIs can be established by UPEC strains with vastly diverse virulence genotypes. We then compared the levels of gene expression of these virulence factors following culture in filter-sterilized urine (*Figure 1B*, *Figure 1—figure supplement 3*) to that during infection. As expected, we detected expression of genes involved in iron acquisition during both in vitro urine culture and human UTI (*Figure 1B*). However, we also observed high strain-to-strain variability in gene expression, especially for *hma*, *iutA*, *iucC* and *fyuA*, which is consistent with previous reports (*Subashchandrabose et al., 2014*).

Most of the adhesin genes were expressed at very low levels both during in vitro culture and infection, with the exception of *fim* genes (*Figure 1B*). Interestingly, we observed high variability in *fim* and *flg* operon expression between patients (*Figure 1C*). In the majority of the cases, we detected high levels of *fim* operon expression (9/14) and low levels of *flg* operon expression (12/14). However, in the sample collected from patient HM07, we observed high levels of both *fim* and *flg* expression, potentially indicating a mixed population of both motile and adherent bacteria present in the sample. Overall, the variability in the expression of adhesin and motility machinery might suggest different stages of infection.

Other virulence factors examined were expressed at either similar or lower levels during human UTI compared to in vitro urine cultures (*Figure 1—figure supplement 3*). Notably, virulence factor carriage varies greatly between UPEC strains and we did not discern any infection-specific gene expression among the virulence factors we examined (*Figure 1—figure supplement 4*).

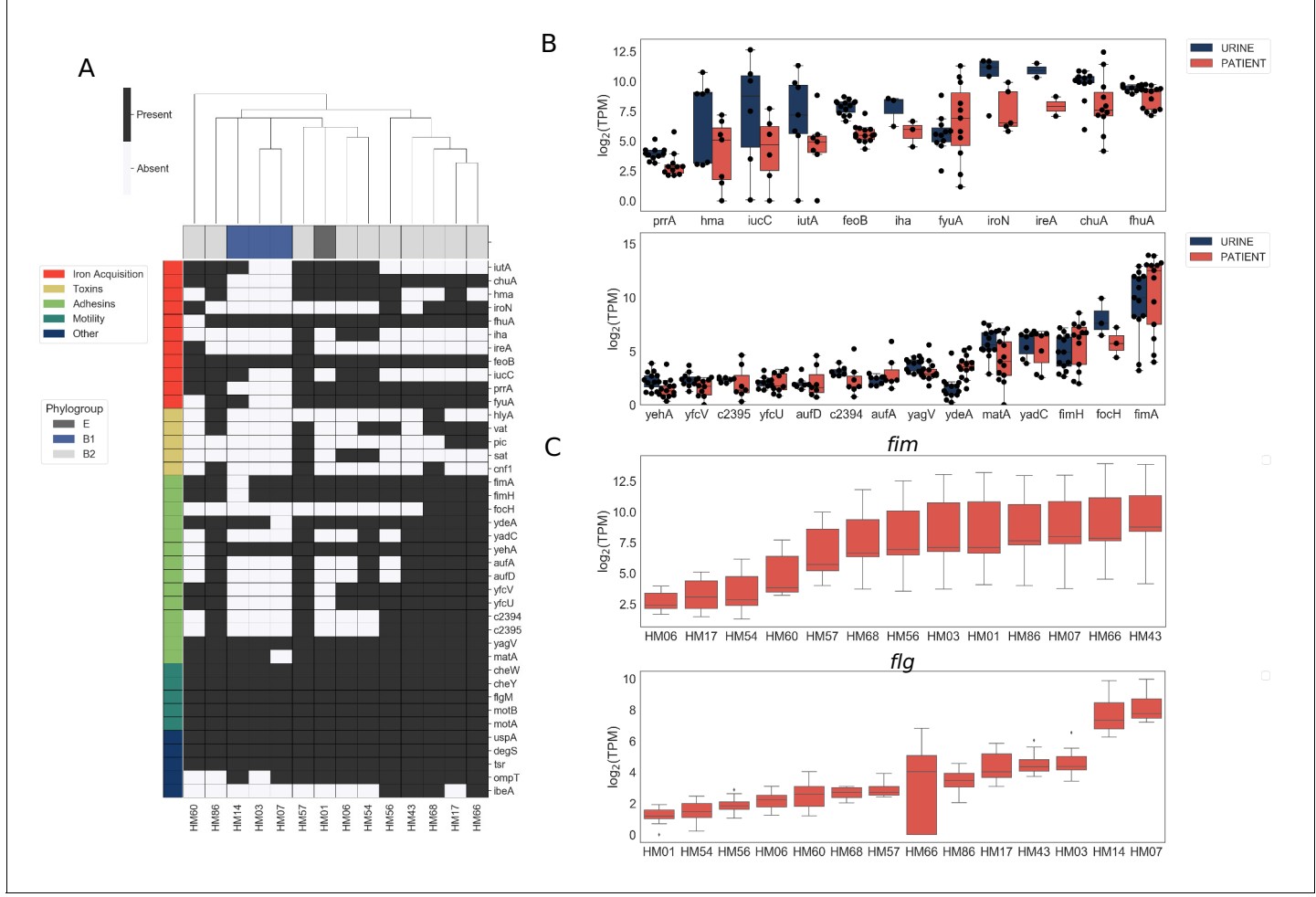

**Figure 1.** Clinical UPEC isolates carry a highly variable set of virulence factors. Phenotypic and genotypic information about the strains can be found in *Figure 1—figure supplement 1*, *Figure 1—figure supplement 2*, *Table 1*, and *Table 2*. (A) Clinical UPEC isolates were examined for presence of 40 virulence factors. Virulence factors were identified based on homology using BLAST searches ($\geq$80% identity,$\geq$90% coverage). The heatmap shows presence (black) or absence (white) of virulence factors across 14 UPEC strains. Hierarchical clustering based on presence/absence of virulence factors shows separate clustering of B1 isolates. (B) Log$_2$ TPM for iron acquisition genes (top panel) and adhesins (bottom panel) in urine and patient samples. Gene expression of other virulence factors is shown in *Figure 1—figure supplement 3*. Correlations of virulence factor expression among in vitro and patient samples is shown in *Figure 1—figure supplement 4*. (C) Log$_2$ TPM of *fim* (top panel) and *flg* (bottom panel) operons across the 14 UPEC strains during in vitro urine culture and human UTI.

DOI: https://doi.org/10.7554/eLife.49748.004

The following figure supplements are available for figure 1:

**Figure supplement 1.** Growth curves for 14 clinical UPEC strains cultured in LB or filter-sterilized urine.
DOI: https://doi.org/10.7554/eLife.49748.005

**Figure supplement 2.** Phylogenetic tree reconstruction of 14 clinical UPEC strains isolated in this study.
DOI: https://doi.org/10.7554/eLife.49748.006

**Figure supplement 3.** Expression of virulence factor genes in urine and patient samples.
DOI: https://doi.org/10.7554/eLife.49748.007

**Figure supplement 4.** Correlations among in vitro and patient samples measured by Pearson correlation coefficient of normalized gene expression of 40 virulence factors plotted according to hierarchical clustering of samples.
DOI: https://doi.org/10.7554/eLife.49748.008

**Figure supplement 5.** Treatment with MICROBEnrich does not affect measures of gene expression.
DOI: https://doi.org/10.7554/eLife.49748.009

## The UPEC core genome exhibits a common gene expression program during clinical infection

Since patient samples contained fewer bacterial reads compared to in vitro controls, we first performed a rigorous quality assurance analysis, which indicated that we possessed sufficient sequencing depth for downstream analyses (*Table 3*, *Table 4*, *Figure 2—figure supplement 1*, *Figure 2—figure supplement 2*, see Materials and methods for details). Next, to perform a comprehensive comparison of gene expression between the different clinical UPEC strains, we identified a set of 2653 genes present in all 14 UPEC strains in this study as well as the reference *E. coli* MG1655 strain (hereafter referred to as the core genome). We then compared the gene expression correlation of the core genome to that of the accessory genome (*i.e.*, 2219 genes that were present in at least two but not all of the clinical UPEC strains) for all 14 isolates cultured in vitro in filter-sterilized urine. As expected for bacterial strains cultured under identical conditions, we saw high correlation of gene expression between any two isolates cultured in vitro irrespective of whether these genes were part of the core or accessory genome (*Figure 2A*). Remarkably, we also observed a high degree of gene expression correlation for the core genome, but not the accessory genome, across all 14 patient samples (*Figure 2B*). This suggested the expression of core genes is conserved during human UTI,

**Table 3.** Summary of alignment statistics (% mapped).

| Sample: | Total reads | Mapped reads | % Mapped | % Mapped to CDS | % Mapped to misc_RNA | % Mapped to rRNA | % Mapped to tRNA | % Mapped to sRNA | % Mapped to tmRNA |
|---|---|---|---|---|---|---|---|---|---|
| HM01 \| UR | 17288419 | 16480326 | 95.3 | 74.91 | 5.51 | 0.01 | 0.26 | 10.2 | 5.49 |
| HM01 \| UTI | 18496607 | 3717040 | 20.1 | 80.44 | 3.36 | 0 | 0.51 | 3.42 | 2.45 |
| HM03 \| UR | 21354719 | 20927541 | 98 | 77.77 | 4.78 | 0 | 0.36 | 9.49 | 5.21 |
| HM03 \| UTI | 16544044 | 8059076 | 48.7 | 80.18 | 2.45 | 0 | 0.86 | 2.23 | 1.35 |
| HM06 \| UR | 23359847 | 22847374 | 97.8 | 78.72 | 3.96 | 0 | 0.33 | 6.3 | 3.23 |
| HM06 \| UTI | 57993519 | 4709092 | 8.1 | 76.94 | 2.62 | 0 | 0.36 | 1.55 | 0.87 |
| HM07 \| UR | 21312224 | 20980473 | 98.4 | 75.2 | 6.02 | 0 | 0.19 | 10.32 | 4.79 |
| HM07 \| UTI | 70804688 | 2097350 | 3 | 73.71 | 4.14 | 0 | 0.6 | 2.08 | 0.77 |
| HM14 \| UR | 21927302 | 21533817 | 98.2 | 76.13 | 5.33 | 0 | 0.15 | 9.97 | 5.16 |
| HM14 \| UTI | 15944762 | 12968218 | 81.3 | 80.51 | 2.21 | 0 | 0.46 | 2.25 | 1.5 |
| HM17 \| UR | 19790215 | 19360294 | 97.8 | 77.41 | 4.29 | 0 | 0.13 | 7.02 | 3.32 |
| HM17 \| UTI | 23874585 | 1842583 | 7.7 | 74.35 | 4.14 | 0 | 0.73 | 2.73 | 1.6 |
| HM43 \| UR | 18541484 | 18239826 | 98.4 | 76.54 | 5.03 | 0 | 0.21 | 9.07 | 4.76 |
| HM43 \| UTI | 58306859 | 8138559 | 14 | 80.38 | 2.76 | 0 | 0.37 | 3.95 | 2.38 |
| HM54 \| UR | 21612581 | 21162544 | 97.9 | 74.96 | 4.13 | 0.01 | 0.12 | 7.17 | 4.06 |
| HM54 \| UTI | 18000843 | 6301998 | 35 | 77.33 | 3.05 | 0.01 | 0.52 | 1.54 | 0.98 |
| HM56 \| UR | 17494135 | 17130847 | 97.9 | 77.93 | 4.09 | 0 | 0.09 | 7.14 | 3.56 |
| HM56 \| UTI | 25408755 | 14935948 | 58.8 | 79.41 | 2.59 | 0 | 0.58 | 1.98 | 1.17 |
| HM57 \| UR | 19253078 | 18966748 | 98.5 | 77.07 | 4.85 | 0 | 0.08 | 8.26 | 3.86 |
| HM57 \| UTI | 105629816 | 926795 | 0.9 | 71.48 | 4.2 | 0 | 0.65 | 2.63 | 1.5 |
| HM60 \| UR | 15898045 | 15651916 | 98.5 | 76.35 | 4.14 | 0 | 0.09 | 7.47 | 4.05 |
| HM60 \| UTI | 76149837 | 764255 | 1 | 70.69 | 3.76 | 0 | 0.7 | 1.84 | 1.04 |
| HM66 \| UR | 17184018 | 16736066 | 97.4 | 74.15 | 4.93 | 0 | 0.12 | 9.53 | 5.28 |
| HM66 \| UTI | 25954183 | 79859 | 0.3 | 65.41 | 2.71 | 0 | 0.46 | 1.42 | 0.67 |
| HM68 \| UR | 15841639 | 15562711 | 98.2 | 78.31 | 2.84 | 0 | 0.14 | 6.03 | 3.67 |
| HM68 \| UTI | 65413931 | 2401089 | 3.7 | 73.11 | 4.8 | 0 | 0.83 | 4.58 | 2.73 |
| HM86 \| UR | 15019669 | 14606346 | 97.2 | 76.06 | 4.09 | 0 | 0.16 | 6.99 | 3.54 |
| HM86 \| UTI | 10667404 | 6413794 | 60.1 | 78.33 | 2.8 | 0 | 0.77 | 3.08 | 1.62 |

DOI: https://doi.org/10.7554/eLife.49748.016

**Table 4.** Summary of alignment statistics (raw counts).

| Sample: | CDS | misc_RNA | rRNA | tRNA | sRNA | tmRNA |
|---|---|---|---|---|---|---|
| HM01 \| UR | 12345933 | 907900 | 1504 | 43435 | 1680592 | 905367 |
| HM01 \| UTI | 2989889 | 124744 | 143 | 19133 | 126985 | 91056 |
| HM03 \| UR | 16274560 | 999727 | 44 | 76181 | 1985885 | 1090263 |
| HM03 \| UTI | 6461781 | 197433 | 24 | 69006 | 179905 | 109081 |
| HM06 \| UR | 17985174 | 904287 | 43 | 76160 | 1439268 | 738927 |
| HM06 \| UTI | 3623181 | 123428 | 23 | 17015 | 72873 | 40864 |
| HM07 \| UR | 15776986 | 1262236 | 177 | 39363 | 2165537 | 1005391 |
| HM07 \| UTI | 1546060 | 86761 | 30 | 12681 | 43708 | 16065 |
| HM14 \| UR | 16393471 | 1148443 | 86 | 32625 | 2146180 | 1110769 |
| HM14 \| UTI | 10441062 | 286490 | 50 | 59823 | 291189 | 194198 |
| HM17 \| UR | 14986237 | 830647 | 48 | 24865 | 1358261 | 642452 |
| HM17 \| UTI | 1370047 | 76227 | 15 | 13494 | 50273 | 29443 |
| HM43 \| UR | 13960276 | 916836 | 21 | 37450 | 1653607 | 867656 |
| HM43 \| UTI | 6541810 | 225003 | 29 | 30200 | 321597 | 194030 |
| HM54 \| UR | 15863933 | 873414 | 1662 | 25326 | 1517844 | 858505 |
| HM54 \| UTI | 4873058 | 192289 | 353 | 32932 | 97321 | 61939 |
| HM56 \| UR | 13349576 | 701313 | 78 | 15697 | 1222601 | 609922 |
| HM56 \| UTI | 11860835 | 386845 | 52 | 86723 | 295607 | 175048 |
| HM57 \| UR | 14617905 | 919256 | 157 | 15069 | 1567276 | 732845 |
| HM57 \| UTI | 662515 | 38910 | 13 | 6057 | 24340 | 13929 |
| HM60 \| UR | 11949731 | 647306 | 62 | 13601 | 1169464 | 633959 |
| HM60 \| UTI | 540215 | 28718 | 11 | 5361 | 14062 | 7958 |
| HM66 \| UR | 12409693 | 825583 | 51 | 19323 | 1595303 | 884439 |
| HM66 \| UTI | 52232 | 2161 | 0 | 366 | 1137 | 534 |
| HM68 \| UR | 12187024 | 442312 | 22 | 22226 | 938831 | 571220 |
| HM68 \| UTI | 1755457 | 115276 | 16 | 19970 | 110052 | 65627 |
| HM86 \| UR | 11110009 | 597368 | 551 | 23424 | 1021292 | 517105 |
| HM86 \| UTI | 5023803 | 179823 | 46 | 49276 | 197828 | 103919 |

DOI: https://doi.org/10.7554/eLife.49748.017

while expression of accessory genome might be more reflective of the specific conditions during each infection. Furthermore, the gene expression correlation within urine samples (*Figure 2C*, *Figure 2D*, median correlation 0.92, URINE:URINE), and within patient samples (*Figure 2C*, *Figure 2D*, median correlation 0.91, PATIENT:PATIENT) was considerably higher than the gene expression correlation between in vitro urine and patient samples (*Figure 2C*, *Figure 2D*, median correlation 0.73, URINE:PATIENT). The gene expression correlation between in vitro and patient samples remained low, even when we directly compared identical strains (*i.e.* HM56 cultured in vitro in urine vs. HM56 isolated from the patient) (*Figure 2C*, *Figure 2D*, median of 0.74, URINE: PATIENT:matched). This analysis suggested that UPEC adopt an infection-specific gene expression program that is distinct from UPEC undergoing exponential growth in urine in vitro. Finally, we independently confirmed this observation using principal component analysis (PCA), which revealed that patient samples form a tight cluster, distinct from in vitro cultures (*Figure 2E*), demonstrating the common transcriptional state of UPEC during human UTI.

We also performed PCA analysis on in vitro (*Figure 2—figure supplement 3A,B*) and patient samples (*Figure 2—figure supplement 3C,D*) separately, to ascertain whether there was any discernible effect of bacterial phylogroup (*Figure 2—figure supplement 3A,C*) or patients' previous history of UTI (*Figure 2—figure supplement 3B,D*) on gene expression. Interestingly, B1 and B2

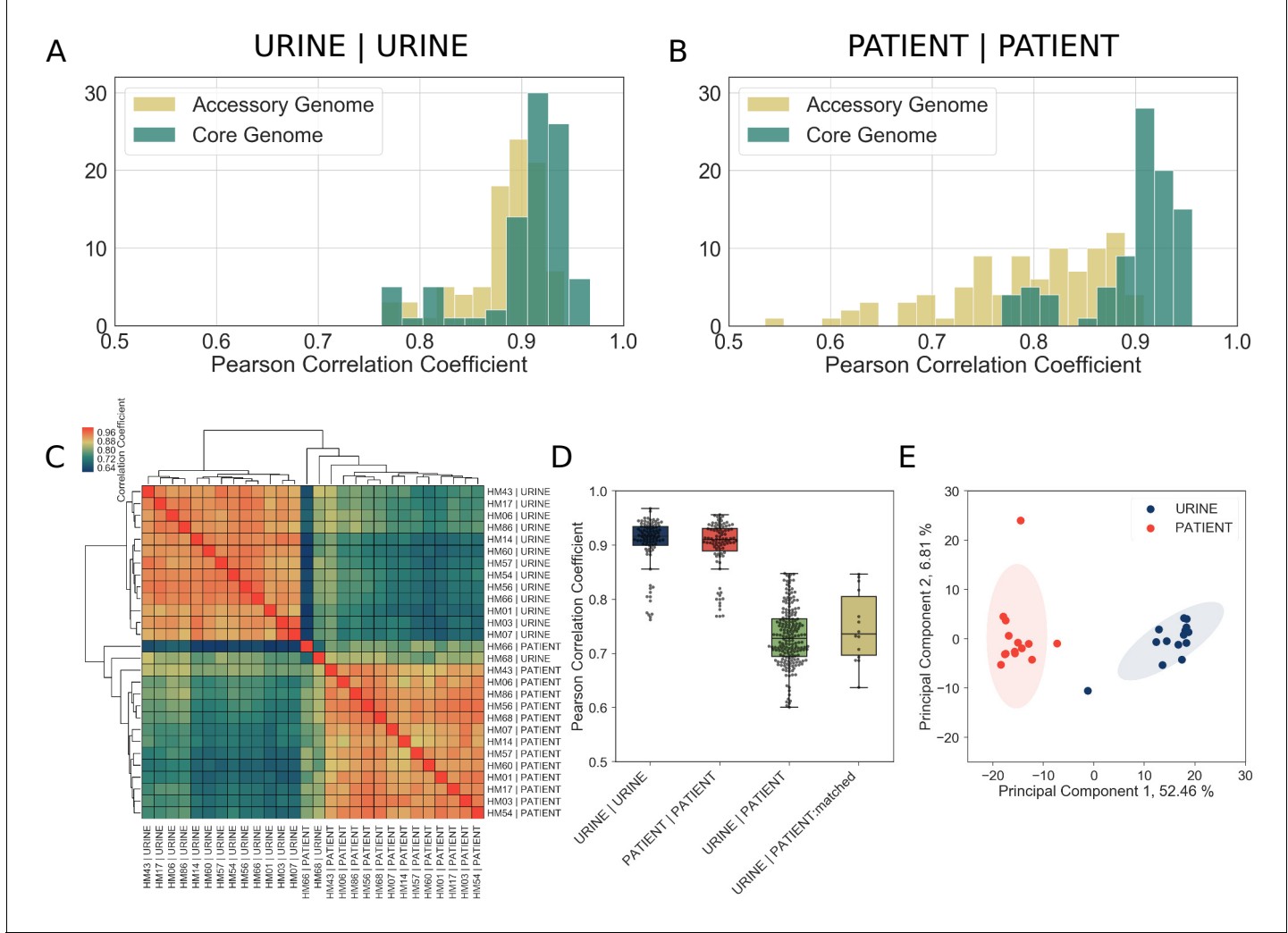

**Figure 2.** Core genome expression in patients is highly correlated. The analysis details are described in Materials and methods, and figure supplements. (**A**)-(**B**) Histogram of Pearson correlation coefficients among all samples cultured in vitro (**A**) or isolated from patients (**B**) based either on core genome or accessory genome comparisons. Accessory genome includes genes that were found in at least two but fewer than 14 of the clinical isolates. (**C**) Correlations among in vitro and patient samples measured by Pearson correlation coefficient of normalized gene expression plotted according to hierarchical clustering of samples. (**D**) Pearson correlation coefficient among all samples cultured in vitro (URINE | URINE, median = 0.92), among all samples isolated from patients (PATIENT | PATIENT, median = 0.91), between samples cultured in urine and samples isolated from patients (URINE | PATIENT, median = 0.73), and between matching urine/patient samples (ex. HM14 | URINE vs HM14 | PATIENT), (URINE | PATIENT:matched, median = 0.74). (**E**) Principal component analysis of normalized gene expression of 14 clinical isolates in patients and in vitro urine cultures shows distinct clustering of in vitro and patient isolates.

DOI: https://doi.org/10.7554/eLife.49748.010

The following source data and figure supplements are available for figure 2:

**Source data 1.** Genes differentially expressed between B1 and B2 phylogroup strains during in vitroculture in urine.
DOI: https://doi.org/10.7554/eLife.49748.014

**Source data 2.** Genes differentially expressed between B1 and B2 phylogroup strains during human UTI.
DOI: https://doi.org/10.7554/eLife.49748.015

**Figure supplement 1.** Saturation curves.
DOI: https://doi.org/10.7554/eLife.49748.011

**Figure supplement 2.** Expression ranges of core genome genes.
DOI: https://doi.org/10.7554/eLife.49748.012

**Figure supplement 3.** Effect of phylogenetic group on core genome expression.
DOI: https://doi.org/10.7554/eLife.49748.013

strains did cluster separately and a number of genes were expressed differentially in B1 and B2 backgrounds (*Figure 2—source data 1*, *Figure 2—source data 2*), suggesting that variation in gene regulatory elements between phylogroups has a small but discernible role in gene expression both in vitro and during infection. However, we found that patients' history of UTI had no effect on bacterial gene expression.

Taken together, our data indicate diverse UPEC strains adopt a specific and conserved transcriptional program for their core genes during human infection.

## UPEC show increased expression of replication and translation machinery during UTI

Differential expression analysis of the infection and in vitro transcriptomes identified 492 differentially expressed genes ($\log_2$ fold change greater than two or less than −2, adjusted *p* values < 0.05) (*Figure 3A*, *Figure 3—source data 1*, *Figure 3—source data 2*). Interestingly, pathway analysis (*Table 5*) and manual curation of the differentially expressed gene list (*Figure 3—source data 1*) revealed that expression of ribosomal subunits (r-proteins), and enzymes involved in rRNA, tRNA modification, purine and pyrimidine metabolism, and ribosome biogenesis are significantly higher in patients compared to in vitro cultures (*Figure 3B*). Together with previous studies (*Bielecki et al., 2014*; *Burnham et al., 2018*; *Forsyth et al., 2018*), these data strongly suggest that replication rates during infection are significantly higher than during mid-exponential growth in urine in vitro.

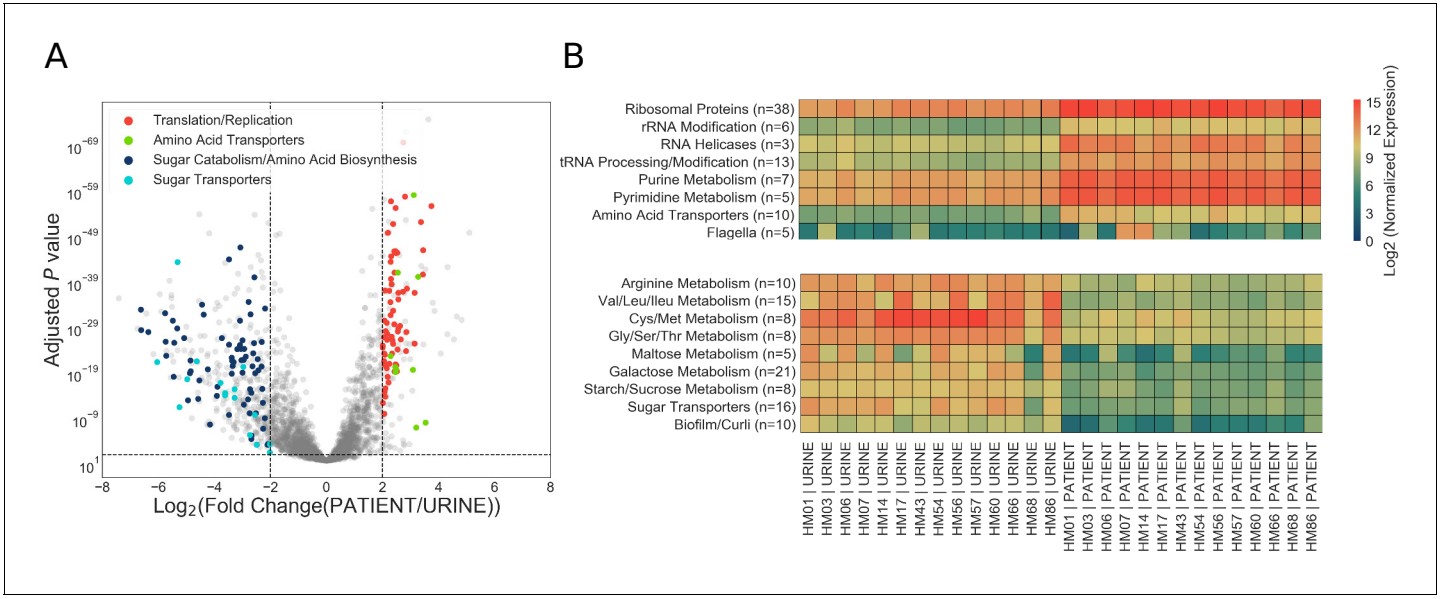

**Figure 3.** Patient-associated transcriptional signature is consistent with rapid bacterial growth. (**A**) The DESeq2 R package was used to compare in vitro urine cultures gene expression to that in patients. Each UPEC strain was considered an independent replicate (n = 14). Genes were considered up-regulated (down-regulated) if $\log_2$ fold change in expression was higher (lower) than 2 (vertical lines), and *P* value < 0.05 (horizontal line). Using these cutoffs, we identified 149 upregulated genes, and 343 downregulated genes. GO/pathway analysis showed that a large proportion of these genes belonged to one of the four functional categories (see legend). For each category, only the genes that have met the significance cut off are shown. The sugar transporters upregulated in UTI patients are shown in figure supplement. (**B**) Mean normalized expression for genes belonging to differentially expressed functional categories/pathways. The number of up or down-regulated genes belonging to each category is indicated next to the category name.

DOI: https://doi.org/10.7554/eLife.49748.019

The following source data and figure supplement are available for figure 3:

**Source data 1.** Genes upregulated during human UTI.
DOI: https://doi.org/10.7554/eLife.49748.021

**Source data 2.** Genes downregulated during human UTI.
DOI: https://doi.org/10.7554/eLife.49748.022

**Figure supplement 1.** Gene expression of four sugar transporters upregulated in UTI patients.
DOI: https://doi.org/10.7554/eLife.49748.020

**Table 5.** GO modules differentially expressed in UTI patients.

| Go id | Annotated | Significant | Expected | P value | Term |
|---|---|---|---|---|---|
| GO:0006518 | 89 | 24 | 16.63 | 0.03134 | peptide metabolic process |
| GO:0016052 | 76 | 36 | 14.2 | 0.00403 | carbohydrate catabolic process |
| GO:0044262 | 75 | 29 | 14.01 | 0.0022 | cellular carbohydrate metabolic process |
| GO:0015980 | 70 | 20 | 13.08 | 0.02632 | energy derivation by oxidation of organic compounds |
| GO:0043043 | 69 | 19 | 12.89 | 0.04306 | peptide biosynthetic process |
| GO:0046395 | 65 | 25 | 12.14 | 0.00556 | carboxylic acid catabolic process |
| GO:0006412 | 63 | 18 | 11.77 | 0.03421 | translation |
| GO:0008643 | 55 | 30 | 10.28 | 0.02488 | carbohydrate transport |
| GO:1903825 | 39 | 12 | 7.29 | 0.04583 | organic acid transmembrane transport |
| GO:0008033 | 38 | 13 | 7.1 | 0.0159 | tRNA processing |
| GO:1905039 | 38 | 12 | 7.1 | 0.03786 | carboxylic acid transmembrane transport |
| GO:0046365 | 38 | 21 | 7.1 | 0.04177 | monosaccharide catabolic process |
| GO:0034219 | 37 | 20 | 6.91 | 0.0005 | carbohydrate transmembrane transport |
| GO:0042710 | 35 | 11 | 6.54 | 0.04746 | biofilm formation |
| GO:0044010 | 34 | 11 | 6.35 | 0.03879 | single-species biofilm formation |
| GO:0006400 | 34 | 11 | 6.35 | 0.03879 | tRNA modification |
| GO:0072329 | 32 | 15 | 5.98 | 0.02795 | monocarboxylic acid catabolic process |
| GO:0009401 | 30 | 11 | 5.6 | 0.01501 | phosphoenolpyruvate-dependent sugar phosphotransferase system |
| GO:0010608 | 29 | 10 | 5.42 | 0.03121 | posttranscriptional regulation of gene expression |
| GO:0034248 | 26 | 9 | 4.86 | 0.03925 | regulation of cellular amide metabolic process |
| GO:0006417 | 26 | 9 | 4.86 | 0.03925 | regulation of translation |
| GO:0015749 | 24 | 13 | 4.48 | 0.03338 | monosaccharide transmembrane transport |
| GO:0051248 | 23 | 9 | 4.3 | 0.01728 | negative regulation of protein metabolic process |
| GO:0044275 | 22 | 11 | 4.11 | 0.04263 | cellular carbohydrate catabolic process |
| GO:0032269 | 22 | 8 | 4.11 | 0.03829 | negative regulation of cellular protein metabolic process |
| GO:0015807 | 19 | 7 | 3.55 | 0.04819 | L-amino acid transport |
| GO:0017148 | 18 | 8 | 3.36 | 0.01044 | negative regulation of translation |
| GO:0034249 | 18 | 8 | 3.36 | 0.01044 | negative regulation of cellular amide metabolic process |
| GO:1902475 | 17 | 7 | 3.18 | 0.02607 | L-alpha-amino acid transmembrane transport |
| GO:0009409 | 14 | 8 | 2.62 | 0.00144 | response to cold |
| GO:0042255 | 14 | 9 | 2.62 | 0.00021 | ribosome assembly |
| GO:0019321 | 14 | 8 | 2.62 | 0.03705 | pentose metabolic process |
| GO:0046835 | 13 | 6 | 2.43 | 0.02143 | carbohydrate phosphorylation |
| GO:0006526 | 12 | 8 | 2.24 | 0.00034 | arginine biosynthetic process |
| GO:0042542 | 10 | 5 | 1.87 | 0.02449 | response to hydrogen peroxide |
| GO:0019323 | 10 | 7 | 1.87 | 0.02539 | pentose catabolic process |

DOI: https://doi.org/10.7554/eLife.49748.018

We also observed infection-specific downregulation of pathways involved in amino acid biosynthesis and sugar metabolism, and a general switch from expression of sugar transporters to that of amino acid transporters (*Figure 3B*, *Figure 3—source data 2*) (with the exception of 4 sugar transporters that were expressed at higher levels in patients: *ptsG*, *fruA*, *fruB*, and *gntU*. *Figure 3—figure supplement 1*). Downregulation of sugar catabolism genes and upregulation of amino acid transporters suggest a metabolic switch to a more specific catabolic program as well as a scavenger lifestyle as elaborated below.

## A shift in metabolic gene expression during UTI to optimize growth potential

During our analysis, we observed that 99% (on average 2621/2653 genes) of core genome was expressed during in vitro culture, in contrast to only 94% in patient samples (2507/2653 genes). Patient samples also contained higher proportion of genes expressed at low levels when compared to in vitro samples. (*Figure 2—figure supplement 2*). Moreover, we noted that the majority of differentially expressed genes were downregulated in patients (343/492 differentially expressed genes). On the other hand, 30% of all upregulated genes (48/149) were ribosomal proteins. Together, these data gave us the first indication that UPEC may undergo a global gene expression reprogramming during urinary tract infection.

Bacterial growth laws postulate that bacteria dedicate a fixed amount of cellular resources to the expression of ribosomes and metabolic machinery. As a consequence, higher growth rates are achieved by allocating resources to ribosome expression at the expense of metabolic machinery production (*Basan, 2018*; *Basan et al., 2015*; *Molenaar et al., 2009*; *Scott et al., 2010*; *Scott and Hwa, 2011*; *You et al., 2013*). However, this resource reallocation between ribosomal and metabolic gene expression has not yet been measured in vivo.

First, we wanted to determine what proportion of the total transcriptome is dedicated to core genome expression. We hypothesized that during infection transcription could shift from the core genome to the accessory genome, which is enriched for virulence factors. However, we found that approximately 50% of total reads mapped to the core genome regardless of whether the bacteria were isolated from the patients or cultured in vitro (*Figure 4A*). Therefore, our data indicated that a fixed proportion of cellular resources were being dedicated to expression of conserved ribosomal and metabolic machinery, regardless of external environment.

We next looked at r-protein expression. Remarkably, we found that almost 25% of core genome reads mapped to r-proteins during infection, while this number was only 7% during exponential growth in urine (*Figure 4B*). These findings support the idea of extremely fast UPEC growth during UTI. Furthermore, this increase in r-protein expression correlated with a marked decrease in the proportion of core genome reads dedicated to the expression of catabolic genes (20% in vitro, 11% in patients, *Figure 4C*) and amino acid biosynthesis genes (5% in vitro, 1% in patients, *Figure 4D*). We then performed the same analysis on our previously published dataset (*Subashchandrabose et al., 2014*), and found a consistent trend of increased r-protein production, and decreased catabolic enzyme expression during human UTI (*Figure 4—figure supplement 1*, *Table 6*, *Table 7*). Thus, our data, which are consistent across multiple data sets, highlight a dramatic and conserved resource reallocation from metabolic gene expression to replication and translational gene expression during human UTI. We postulate that this resource reallocation is required to facilitate the rapid growth rate of UPEC in the host, which has been previously documented (*Burnham et al., 2018*; *Forsyth et al., 2018*).

## Increase in r-protein transcripts is an infection-specific response

Doubling time during exponential growth in urine is longer than the doubling time during exponential growth in rich media, such as LB (*Plank and Harvey, 1979*). Thus, we wanted to determine whether the differences between the infection-specific and in vitro transcriptomes are due to longer doubling times of UPEC cultured in urine. For that purpose, one of the clinical strains, HM43, was cultured in LB, and in a new batch of filter sterilized urine. Using the growth curves shown in *Figure 5A*, we estimated the doubling time of HM43 during exponential growth in LB to be approximately 33 min and the doubling time in urine to be 54 min. In addition, we sequenced RNA from 3-

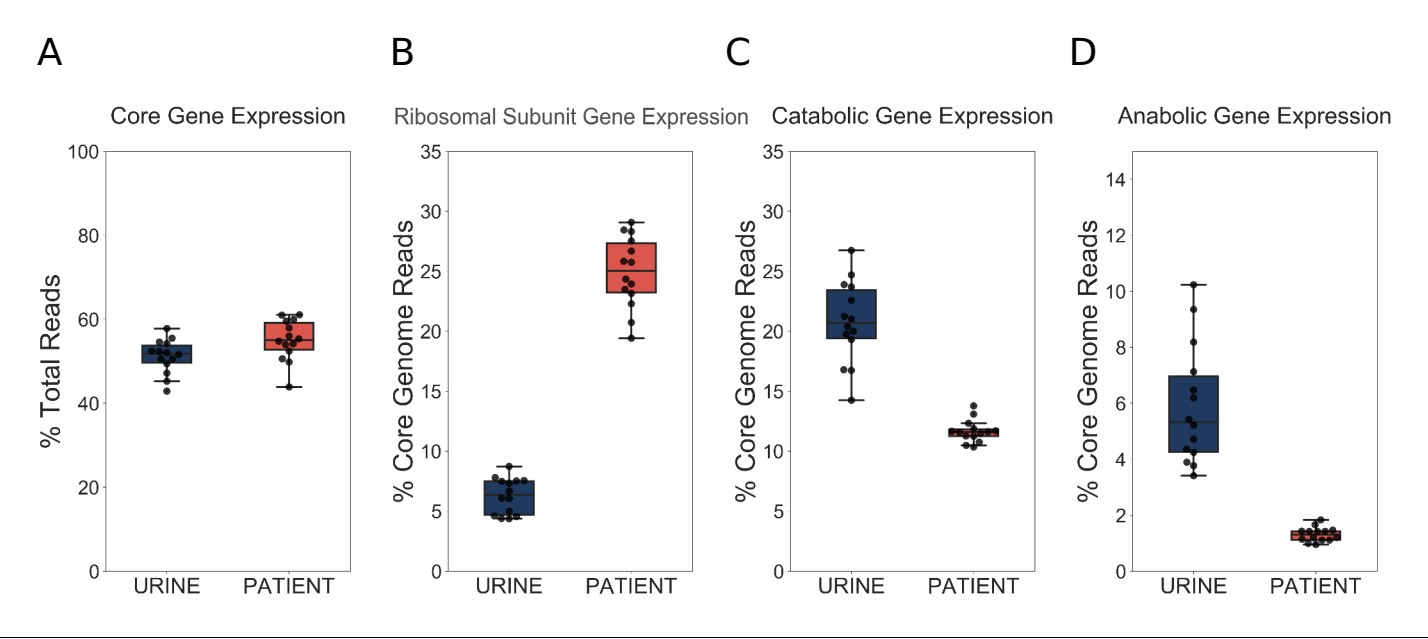

**Figure 4.** UPEC optimize growth potential via resource reallocation during UTI. (**A**) Percentage of reads that aligned to the core genome (2653 genes) out of total mapped reads. (**B**) Percentage of core genome reads that mapped to r-proteins (ribosomal subunit proteins, 48 genes). (**C**) Percentage of core genome reads that mapped to catabolic genes (defined as genes regulated by Crp and present in the core genome (277 genes). (**D**) Percentage of core genome reads that mapped to amino acid biosynthesis genes (54 genes). The equivalent analysis of *Subashchandrabose et al. (2014)* dataset is shown in the figure supplement.

DOI: https://doi.org/10.7554/eLife.49748.023

The following figure supplement is available for figure 4:

**Figure supplement 1.** Resource reallocation analysis of *Subashchandrabose et al. (2014)* dataset.
DOI: https://doi.org/10.7554/eLife.49748.024

hour-old LB cultures, 3-hour-old urine cultures and from the urine of CBA/J mice, 48 hr after trans-urethral inoculation with HM43 (*Table 8*, *Table 9*).

We then determined the proportion of r-protein transcripts in the HM43 transcriptomes isolated from urine and LB cultures. Consistent with our previous experiments, this proportion was very small in urine culture (4%). Interestingly, while the proportion of r-protein transcripts was approximately three times larger in LB cultures compared to urine, it was still significantly lower compared to what we observed during infection (*Figure 5B*). In contrast, the bacterial transcriptome during mouse infection exhibited r-protein expression that was similar to the human infection (*Figure 5B*). Additionally, the proportion of the transcriptome dedicated to catabolic gene expression was highest during urine cultures and lowest during mouse and human infections, indicating a negative correlation between the expression of r-protein and sugar catabolism genes. (*Figure 5C*). Overall, we show that exponential growth in rich medium alone cannot recapitulate the transcriptional signature observed during human infection. Taken together, our data suggest that the resource reallocation described in this study is an infection-specific response.

## Environment-responsive regulators facilitate patient-specific gene expression program

We next sought to identify potential regulators involved in resource reallocation that facilitate the infection-specific UPEC gene expression program. To do so, we performed gene set enrichment analysis (GSEA) on *E. coli* co-regulated genes (regulons). This analysis allowed us to identify regulons enriched in differentially expressed genes. We identified 22 transcriptional factors whose regulon's expression was statistically different between infection and in vitro cultures (*Table 10*). 18/22 regulons were expressed at higher level during in vitro culture, and eight representative regulons are shown in *Figure 6*. Overall, we found that these regulons accounted for 50% of differentially

**Table 6.** Summary of alignment statistics (% mapped) for *Subashchandrabose et al. (2014)*.

| Sample: | Total | Mapped reads | % Mapped | Mapped to CDS | Mapped to misc_RNA | Mapped to rRNA | Mapped to tRNA | Mapped to tmRNA |
|---|---|---|---|---|---|---|---|---|
| HM46 \| UR | 84195438 | 81447525 | 96.74 | 2.41 | 0.05 | 60.55 | 0.01 | 0.01 |
| HM26 \| UTI | 20253252 | 1000968 | 4.94 | 16.75 | 0.24 | 21.24 | 0.09 | 0.16 |
| HM46 \| UTI | 63338418 | 10783798 | 17.03 | 6.93 | 0.12 | 40.3 | 0.1 | 0.1 |
| HM27 \| LB | 67422498 | 65065615 | 96.5 | 2.25 | 0.04 | 55.6 | 0.02 | 0.01 |
| HM27 \| UTI | 67258748 | 18308171 | 27.22 | 9.25 | 0.13 | 45.49 | 0.08 | 0.2 |
| HM26 \| UR | 62242978 | 59994538 | 96.39 | 2.31 | 0.08 | 60.58 | 0.01 | 0.01 |
| HM65 \| LB | 73451346 | 71221338 | 96.96 | 2.53 | 0 | 51.41 | 0.01 | 0 |
| HM69 \| LB | 137690758 | 133649727 | 97.07 | 3.49 | 0.05 | 67.26 | 0.01 | 0.01 |
| HM69 \| UTI | 72509214 | 38506559 | 53.11 | 6.52 | 0.13 | 42.09 | 0.04 | 0.21 |
| HM46 \| LB | 78018026 | 75590297 | 96.89 | 2.78 | 0.06 | 56.9 | 0.01 | 0.01 |
| HM27 \| UR | 98185180 | 94683534 | 96.43 | 2.82 | 0.03 | 61 | 0.01 | 0.01 |
| HM26 \| LB | 70919896 | 68671798 | 96.83 | 2.02 | 0.06 | 55.74 | 0.02 | 0.01 |
| HM65 \| UR | 76024008 | 73555939 | 96.75 | 2.49 | 0 | 55.04 | 0.01 | 0 |
| HM65 \| UTI | 73446576 | 59696718 | 81.28 | 6.19 | 0 | 40.3 | 0.04 | 0 |
| HM69 \| UR | 67112750 | 64834311 | 96.61 | 2.45 | 0.04 | 52.92 | 0.01 | 0.01 |

DOI: https://doi.org/10.7554/eLife.49748.025

expressed genes that were determined to be significantly down-regulated. In contrast, only 6% of upregulated genes belonged to the four regulons that were expressed at higher levels during infection. These included genes involved in the SOS response, as well as purine synthesis (*Table 10*).

**Table 7.** Summary of alignment statistics (% mapped) for *Subashchandrabose et al. (2014)*.

| Sample | CDS | misc_RNA | rRNA | tRNA | tmRNA |
|---|---|---|---|---|---|
| HM46 \| UR | 1960841 | 36901 | 49312604 | 7302 | 5604 |
| HM26 \| UTI | 167663 | 2366 | 212641 | 949 | 1605 |
| HM46 \| UTI | 747702 | 12948 | 4345881 | 10289 | 11281 |
| HM27 \| LB | 1463627 | 26081 | 36173268 | 11717 | 5088 |
| HM27 \| UTI | 1693448 | 24245 | 8329004 | 14427 | 36287 |
| HM26 \| UR | 1387110 | 48847 | 36345620 | 6532 | 5837 |
| HM65 \| LB | 1801858 | 0 | 36612190 | 7263 | 1 |
| HM69 \| LB | 4664579 | 71881 | 89896218 | 13828 | 7949 |
| HM69 \| UTI | 2511733 | 51962 | 16206680 | 17070 | 81355 |
| HM46 \| LB | 2099493 | 42356 | 43011663 | 11135 | 8549 |
| HM27 \| UR | 2673283 | 31185 | 57757240 | 10152 | 8399 |
| HM26 \| LB | 1385766 | 38971 | 38278745 | 11081 | 5724 |
| HM65 \| UR | 1828039 | 0 | 40486611 | 5675 | 1 |
| HM65 \| UTI | 3697360 | 0 | 24059705 | 24055 | 2 |
| HM69 \| UR | 1587484 | 26322 | 34308170 | 4737 | 7686 |

DOI: https://doi.org/10.7554/eLife.49748.026

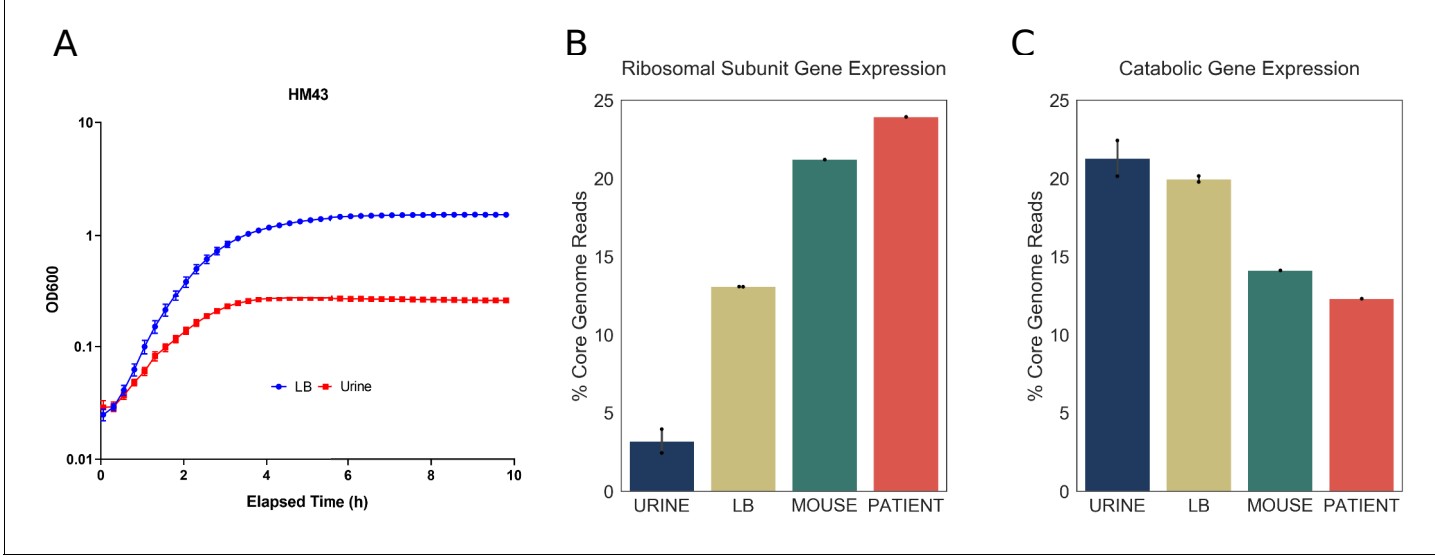

**Figure 5.** Increased expression of ribosomal subunit transcripts is a host specific response. (A) Growth curve for HM43 strain cultured in LB and filter-sterilized urine. (B) Percentage of HM43 core genome reads that mapped to ribosomal subunit proteins under different conditions (URINE: in vitro culture in filter-sterilized urine, LB: in vitro culture in LB, MOUSE: mice with UTI, PATIENT: human UTI. (C) Percentage of HM43 core genome reads that mapped to catabolic genes under different conditions.

DOI: https://doi.org/10.7554/eLife.49748.027

In support of our previous data, the expression of catabolic genes controlled by the Crp regulator was lower in patients compared to urine cultures. In conjunction with the previously described role for Crp in resource reallocation (*You et al., 2013*), our in vivo findings strongly suggest that catabolite repression plays an important role in bacterial growth rate during UTI. Interestingly, other regulators identified in this analysis (NarL, ModE, MetJ, GadE, YdeO) are known sensors of environmental cues, suggesting that the infection-specific gene expression program may be driven by additional environmental signals. Taken together, we propose a model where simultaneous sensing of multiple environmental cues in the urinary tract leads to the global down-regulation of multiple metabolic regulons during infection. The cellular resources (*e.g.*, RNA polymerase) that are freed as a result are then allocated to the transcription of genes (for example, r-proteins), which are required to maintain rapid growth rate.

## Discussion

UPEC causes one of the most prevalent bacterial infections in humans; consequently, the virulence mechanisms of UPEC infection have been well-characterized. However, while we know that these

**Table 8.** Summary of alignment statistics (% mapped) for mouse UTI study.

| Sample | Total reads | Mapped reads | % Mapped | Mapped to CDS | Mapped to misc_RNA | Mapped to rRNA | Mapped to tRNA | Mapped to sRNA | Mapped to tmRNA |
|---|---|---|---|---|---|---|---|---|---|
| HM43 \| LB \| rep1 | 63966646 | 62813946 | 98.2 | 73.01 | 5.49 | 0 | 0.2 | 11.03 | 6.41 |
| HM43 \| LB \| rep2 | 37833957 | 37090863 | 98.04 | 71.59 | 5.91 | 0 | 0.2 | 11.63 | 6.69 |
| HM43 \| UR \| rep1 | 43179946 | 42293006 | 97.95 | 63 | 8.9 | 0 | 0.06 | 19.96 | 11.94 |
| HM43 \| UR \| rep2 | 44176952 | 43093840 | 97.55 | 53.64 | 10.94 | 0.01 | 0.03 | 27.8 | 17.9 |
| HM43 \| mouse | 44314537 | 3690174 | 8.33 | 76.72 | 2.75 | 0 | 0.24 | 6.11 | 4 |

DOI: https://doi.org/10.7554/eLife.49748.028

**Table 9.** Summary of alignment statistics (% mapped) for mouse UTI study.

| Sample | CDS | misc_RNA | rRNA | tRNA | sRNA | tmRNA |
|---|---|---|---|---|---|---|
| HM43 \| LB \| rep1 | 45862961 | 3449232 | 327 | 123950 | 6929261 | 4028787 |
| HM43 \| LB \| rep2 | 26554546 | 2192539 | 204 | 74396 | 4312075 | 2482416 |
| HM43 \| UR \| rep1 | 26644071 | 3765281 | 218 | 26488 | 8439668 | 5049595 |
| HM43 \| UR \| rep2 | 23115456 | 4714597 | 2962 | 14049 | 11979913 | 7714978 |
| HM43 \| mouse | 2831120 | 101419 | 55 | 8994 | 225533 | 147467 |

DOI: https://doi.org/10.7554/eLife.49748.029

virulence strategies (*e.g.*, iron acquisition, adhesion, immune evasion) are essential for establishing infection, UPEC strains can differ dramatically in the specific factors that are utilized. Additionally, our data indicate that the expression of virulence factors can change from patient to patient, suggesting that the need for a specific factor might vary during the course of the infection.

In this study, we set out to uncover universal bacterial features during human UTIs, regardless of the stage of the infection or patient history. To do so, we performed transcriptomic analysis on bacterial RNA isolated directly from the urine of 14 patients and compared it to the gene expression of

**Table 10.** GSEA results.
Gene sets found to be enriched in differentially expressed genes. For example, Lrp, Repressor indicates gene set repressed by Lrp (data obtained from RegulonDB 9.4). Expression indicates whether regulon expression was higher in patients of during in vitro culture in urine. Regulon size: number of genes in the gene set; Matched size: number of genes found in data set; NES: normalized enrichment score; FDR: false discovery rate.

| | Function | Expression (higher in) | Regulon size | Matched size | NES | FDR |
|---|---|---|---|---|---|---|
| Lrp | Repressor | Urine | 85 | 27 | 2.29079978 | 0 |
| NarL | Repressor | Urine | 87 | 65 | 2.24435801 | 0 |
| Lrp | Activator | Urine | 38 | 19 | 2.21269565 | 0 |
| MetJ | Repressor | Urine | 15 | 14 | 2.12885223 | 0.00083422 |
| Crp | Activator | Urine | 425 | 277 | 2.12150402 | 0.00066738 |
| CsgD | Activator | Urine | 13 | 12 | 2.01197693 | 0.00250267 |
| GadX | Activator | Urine | 23 | 15 | 1.89350304 | 0.00929563 |
| ModE | Activator | Urine | 31 | 28 | 1.87289606 | 0.0108449 |
| YdeO | Activator | Urine | 18 | 14 | 1.81975146 | 0.02002136 |
| Fur | Repressor | Urine | 110 | 66 | 1.76658693 | 0.02752936 |
| PhoP | Activator | Urine | 45 | 33 | 1.7607379 | 0.0256334 |
| RcsB | Activator | Urine | 58 | 28 | 1.70667558 | 0.03781812 |
| Hns | Repressor | Urine | 144 | 62 | 1.69880665 | 0.03657748 |
| GadE | Activator | Urine | 70 | 38 | 1.69400478 | 0.03515655 |
| RcsA | Activator | Urine | 42 | 24 | 1.68615633 | 0.03448122 |
| NarP | Activator | Urine | 32 | 29 | 1.65675898 | 0.04045982 |
| NarP | Repressor | Urine | 33 | 26 | 1.6406359 | 0.04279074 |
| FhlA | Activator | Urine | 30 | 15 | 1.62536048 | 0.04514074 |
| FliZ | Repressor | Urine | 20 | 15 | 1.60948953 | 0.04750681 |
| LexA | Repressor | Patients | 59 | 43 | −1.696072 | 0.03586007 |
| Cra | Repressor | Patients | 59 | 50 | −1.7121855 | 0.04267527 |
| PurR | Repressor | Patients | 31 | 31 | −1.752299 | 0.04410253 |
| FadR | Activator | Patients | 12 | 11 | −1.9871524 | 0.00342544 |

DOI: https://doi.org/10.7554/eLife.49748.031

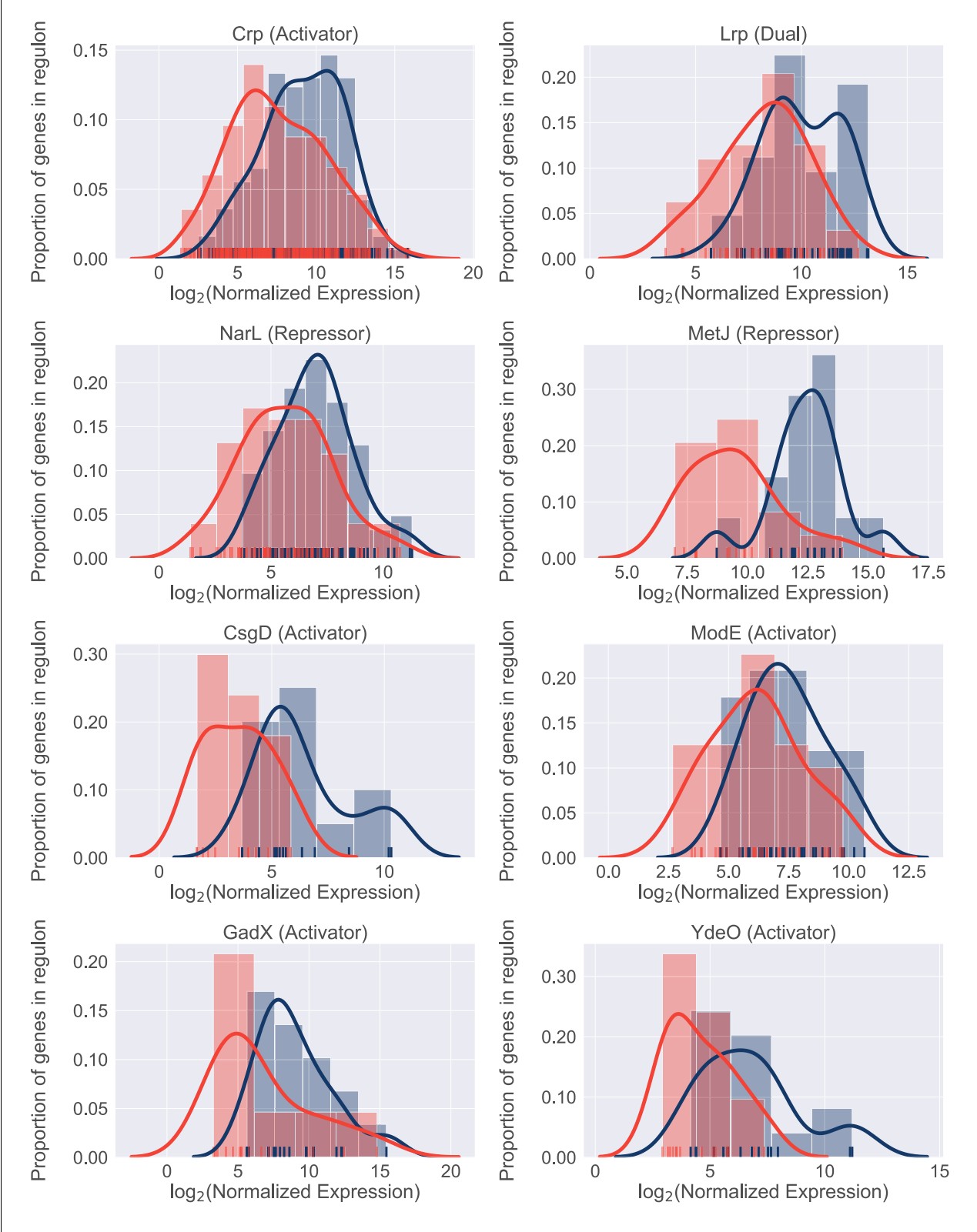

**Figure 6.** Differential regulon expression suggests role for multiple regulators in resource reallocation. Regulon expression for 8 out of 22 regulons enriched for genes downregulated in the patients. Expression of each gene in the regulon during in vitro culture (blue) or during UTI (red) is shown along the x-axis. Histograms show proportion of genes in the regulon expressed at any given level.
DOI: https://doi.org/10.7554/eLife.49748.030

identical strains cultured to mid-exponential phase in sterile urine. Our analysis focused on the core genome as opposed to the more commonly studied accessory genome, which contains the majority of the classical virulence factors. This allowed us to identify a remarkably conserved gene expression signature shared by all 14 UPEC strains during UTI.

Although frequently overlooked, bacterial metabolism is an essential component of bacterial pathogenesis. Since the core genome is enriched for metabolic genes, we anticipated that our study would illuminate the UPEC metabolic state during human infection. Our data revealed an infection-specific increase in ribosomal protein expression in all 14 UPEC isolates, which was suggestive of bacteria undergoing rapid growth. These data strongly support the previous findings of *Bielecki et al. (2014)*, which found a gene expression profile consistent with rapid growth in elderly patients with UTIs. Furthermore, while we did observe increased r-protein expression in exponentially growing UPEC cultured in LB, these transcripts were dramatically more abundant in the context of infection (human and mouse). Thus, the findings that UPEC maintain a conserved gene expression during UTI and grow faster in the host in comparison to in vitro conditions is consistent across multiple studies and patient cohorts (*Bielecki et al., 2014*), and supports recent studies that have documented very rapid UPEC growth rate measured directly in patients (*Burnham et al., 2018*; *Forsyth et al., 2018*).

Importantly, our analysis reveals how this growth rate can be achieved. We found that regardless of external environment,~50% of total gene expression is allocated to the core genome, consisting of metabolic and replication machinery, which mediate bacterial growth potential. When the infection-specific transcriptome was compared to that of UPEC cultured to mid-exponential phase in urine, we observed that elevated levels of ribosomal transcripts correlated with decreased levels of metabolic gene expression. We propose that this reallocation of resources within the core genome drives the rapid growth rate of UPEC during infection.

This resource reallocation is equivalent to what has been described as the bacterial 'growth law'. Based on in vitro studies, the growth law proposes that increases in ribosomal gene expression occurs at the expense of a cell's metabolic gene expression (*Basan, 2018*; *Scott et al., 2010*). Our analysis of UPEC gene expression directly from patients is consistent with this hypothesis. In addition, regulatory network analysis revealed that multiple metabolic regulons exhibit decreased transcript levels in patients suggesting an actively regulated process. In contrast, synthesis of ribosomal RNA (rRNA) coordinates the expression of ribosomal proteins by a translational feedback regulation mechanism (*Jin et al., 2012*; *Jin and Cabrera, 2006*; *Nomura et al., 1984*). rRNA synthesis is proposed to be regulated by the competition of RNA polymerase between transcription of rRNA operons and that of other genes, with some studies suggesting that mid-log growing cells might require almost all RNA polymerase dedicated to rRNA synthesis (*Jin et al., 2012*; *Jin and Cabrera, 2006*). Thus, decreased metabolic gene expression could allow the cell to shift its allocation of RNA polymerase towards rRNA synthesis and as a result, ribosomal protein expression. Although we cannot exclude other mechanisms, we propose that the reallocation of RNA polymerase molecules from metabolic genes to rRNA and ribosomal protein genes is a common feature adopted by diverse UPEC to promote rapid growth during UTI.

Three recent studies have attempted to characterize UPEC gene expression in patients with UTIs (*Bielecki et al., 2014*; *Hagan et al., 2010*; *Subashchandrabose et al., 2014*). These studies focused on the importance of virulence factor expression in specific strains and have demonstrated changes in gene expression between infection and in vitro cultures. It should be noted that all of these studies, as well as our own, were performed using bacterial RNA isolated from patient urine (that was immediately stabilized upon collection). As a result, we cannot exclude the possibility that gene expression of UPEC residing in the bladder may differ from UPEC isolated from patient urine. However, the fact remains that we and others (*Bielecki et al., 2014*) report that patients with different histories of UTIs all harbored a population of actively dividing bacteria in a remarkably specific metabolic state, which we have also recapitulated in a mouse model of infection in this study.

These findings raise a number of interesting questions. Firstly, how is rapid growth rate beneficial to UPEC? For example, rapid growth rate could be necessary to avoid the hosts' innate immune response such as micturition or epithelial cell shedding. Additionally, how does this growth rate influence the tempo and mode of bacterial evolution, especially with regards to genomic integrity and the acquisition of antibiotic resistance? Finally, what are the external cues that launch the infection-specific transcriptional response? It has been noted previously that filtered urine lacks some proteins

that are present in unfiltered urine (*Greene et al., 2015*), thus it would be interesting to see if supplementation of filtered urine with specific proteins/metabolites could recapitulate in vivo phenotype. While our study was not designed to identify infection-specific metabolites, our regulatory network analysis suggests that multiple environmental cues might reinforce the suppression of metabolic gene expression. We suggest that identifying and targeting these environmental cues is a promising approach to limit UPEC growth during UTI and gain the upper hand on this pathogen.

# Materials and methods

**Key resources table**

| Reagent type (species) or resource | Designation | Source or reference | Identifiers | Additional information |
|---|---|---|---|---|
| Strain, strain background (*Escherichia coli*) | Uropathogenic *Escherichia coli* HM01 | This study | | Strain isolation described in Study Design section below |
| Strain, strain background (*Escherichia coli*) | Uropathogenic *Escherichia coli* HM03 | This study | | Strain isolation described in Study Design section below |
| Strain, strain background (*Escherichia coli*) | Uropathogenic *Escherichia coli* HM06 | This study | | Strain isolation described in Study Design section below |
| Strain, strain background (*Escherichia coli*) | Uropathogenic *Escherichia coli* HM07 | This study | | Strain isolation described in Study Design section below |
| Strain, strain background (*Escherichia coli*) | Uropathogenic *Escherichia coli* HM14 | This study | | Strain isolation described in Study Design section below |
| Strain, strain background (*Escherichia coli*) | Uropathogenic *Escherichia coli* HM17 | This study | | Strain isolation described in Study Design section below |
| Strain, strain background (*Escherichia coli*) | Uropathogenic *Escherichia coli* HM43 | This study | | Strain isolation described in Study Design section below |
| Strain, strain background (*Escherichia coli*) | Uropathogenic *Escherichia coli* HM54 | This study | | Strain isolation described in Study Design section below |
| Strain, strain background (*Escherichia coli*) | Uropathogenic *Escherichia coli* HM56 | This study | | Strain isolation described in Study Design section below |
| Strain, strain background (*Escherichia coli*) | Uropathogenic *Escherichia coli* HM57 | This study | | Strain isolation described in Study Design section below |
| Strain, strain background (*Escherichia coli*) | Uropathogenic *Escherichia coli* HM60 | This study | | Strain isolation described in Study Design section below |
| Strain, strain background (*Escherichia coli*) | Uropathogenic *Escherichia coli* HM66 | This study | | Strain isolation described in Study Design section below |
| Strain, strain background (*Escherichia coli*) | Uropathogenic *Escherichia coli* HM68 | This study | | Strain isolation described in Study Design section below |
| Strain, strain background (*Escherichia coli*) | Uropathogenic *Escherichia coli* HM86 | This study | | Strain isolation described in Study Design section below |
| Strain, strain background (*Escherichia coli*) | Uropathogenic *Escherichia coli* HM26 | (*Subashchandrabose et al., 2014*) | | |

*Continued on next page*

*Continued*

| Reagent type (species) or resource | Designation | Source or reference | Identifiers | Additional information |
|---|---|---|---|---|
| Strain, strain background (*Escherichia coli*) | Uropathogenic *Escherichia coli* HM27 | (*Subashchandrabose et al., 2014*) | | |
| Strain, strain background (*Escherichia coli*) | Uropathogenic *Escherichia coli* HM46 | (*Subashchandrabose et al., 2014*) | | |
| Strain, strain background (*Escherichia coli*) | Uropathogenic *Escherichia coli* HM65 | (*Subashchandrabose et al., 2014*) | | |
| Strain, strain background (*Escherichia coli*) | Uropathogenic *Escherichia coli* HM69 | (*Subashchandrabose et al., 2014*) | | |
| Strain, strain background (*Mus musculus*) | CBA/J | | | |
| commercial assay or kit | MICROBEnrich Kit | Thermo Fisher | AM1901 | |
| commercial assay or kit | RNeasy kit | Qiagen | 74104 | |
| commercial assay or kit | Turbo DNase kit | Thermo Fisher | AM2238 | |
| commercial assay or kit | iScript cDNA synthesis kit | Bio Rad | 1708890 | |
| commercial assay or kit | ScriptSeq Complete Gold Kit (Epidemiology) | Illumina | Discontinued | |
| commercial assay or kit | ScriptSeq Complete Kit (Bacteria) | Illumina | Discontinued | |
| commercial assay or kit | PowerUP SYBR Green Master Mix | Bio Rad | A25779 | |
| commercial assay or kit | Dynabeads mRNA DIRECT Purification kit | Thermo Fisher | 61011 | |
| chemical compound, drug | RNAprotect | Qiagen | 76526 | |
| software, algorithm | Trimmomatic | (*Bolger et al., 2014*) | 0.36 | |
| software, algorithm | Bowtie2 | (*Langmead and Salzberg, 2012*) | 2.3.4 | |
| software, algorithm | samtools | (*Li, 2011*) | 1.5 | |
| software, algorithm | HTseq | (*Anders et al., 2015*) | 0.9.1 | |
| software, algorithm | Get_homologues | (*Contreras-Moreira and Vinuesa, 2013*) | 20170807 | |
| software, algorithm | DESeq2 | (*Love et al., 2014*) | 1.22.2 | |

## Study design

Sample collection was previously described (*Subashchandrabose et al., 2014*). Briefly, a total of 86 female participants, presenting with symptoms of lower UTI at the University of Michigan Health Service Clinic in Ann Arbor, MI in 2012, were enrolled in this study. The participants were compensated with a $10 gift card to a popular retail store. Clean catch midstream urine samples from participants

were immediately stabilized with two volumes of RNAprotect (Qiagen) to preserve the in vivo transcriptional profile. De-identified patient samples were assigned unique sample numbers and used in this study. Of the 86 participants, 38 were diagnosed with UPEC-associated UTIs (Subashchandrabose et al., 2014). Of these, 19 samples gave us sufficient RNA yield of satisfactory quality. Five were used for a pilot project (Subashchandrabose et al., 2014), the remaining 14 were used in this study.

## Genome sequencing and assembly

The genomic DNA from clinical strains of *E. coli* were isolated with CTAB/phenol-chloroform based protocol. Library preparation and sequencing were performed on PacBio RS system at University of Michigan Sequencing Core. De novo assemblies were performed with canu de novo assembler (Koren et al., 2017) with all the parameters set to default mode and correction phase turned on. Finished genome assembly of reference strains (MG1655, CFT073, UTI89, EC958) were downloaded from NCBI and were converted to fastq reads using ArtificialFastqGenerator v1.0. Trimmomatic 0.36 (Bolger et al., 2014) was used for trimming adapter sequences. Variants were identified by (i) mapping filtered reads to reference genome sequence CFT073 (NC_004431) using the Burrows-Wheeler short-read aligner (bwa-0.7.17) (Li and Durbin, 2009), (ii) discarding polymerase chain reaction duplicates with Picard (picard-tools-2.5.0), and (iii) calling variants with SAMtools (samtools-1.2,) (Li, 2011) and bcftools (Li, 2011). Variants were filtered from raw results using GATK' s (GenomeAnalysisTK-3.3–0 [Van der Auwera et al., 2013]) VariantFiltration (QUAL,>100; MQ,>50; DP >= 10 reads supporting variant; and FQ <0.025). In addition, a custom python script was used to filter out single-nucleotide variants that were <5 base pairs (bp) in proximity to indels. Positions that fell under the following regions were masked (substituted with N): (i) Phage and Repeat region of the reference genome (identified using Phaster and Nucmer; MUMmer3.23 [Kurtz et al., 2004]) (ii) Low MQ and Low FQ regions (ii) base positions that didn't pass the hard filters (QUAL,>100; DP >= 10) were individually masked in each sample. Recombinant region identified by Gubbins 2.3.1 (Croucher et al., 2015) were filtered out and a maximum likelihood tree was constructed in RAxML 8.2.8 (Stamatakis, 2014) using a general-time reversible model of sequence evolution from the gubbins filtered alignment. Bootstrap analysis was performed with the number of bootstrap replicates determined using the bootstrap convergence test and the autoMRE convergence criteria (-N autoMRE). Bootstrap support values were overlaid on the best scoring tree identified during rapid bootstrap analysis (-f a).

## Phylogroup, MLST, and serogroup typing

Phylogroups were assigned using an in-house script based on the presence and absence of primer target sequences and typing scheme (Clermont et al., 2013). MLST schemes from pubmlst (Jolley et al., 2018) were downloaded using ARIBA's pubmlstget tool and sequence types were determined by running ARIBA (Hunt et al., 2017) against this pubmlst database. Serogroups were determined using SerotypeFinder (Joensen et al., 2015).

## Bacterial culture conditions

Human urine was pooled from four age-matched healthy female volunteers. Overnight cultures of clinical isolates were washed once in human urine, then 250 µl of overnight culture was added to 25 ml of filter-sterilized human urine and cultured statically at 37C for 2 hours. Six milliliters of this culture were stabilized with RNAprotect (Qiagen) and used for RNA purification.

## Antibiotic resistance profiling

As described in Subashchandrabose et al. (2014), identity and antibiotic resistance profiles of UPEC isolates were determined using a VITEK2 system (BioMerieux).

## RNA isolation and sequencing

RNA isolation protocol was previously described (Subashchandrabose et al., 2014). Briefly, samples were treated with proteinase K and total RNA was isolated using Qiagen RNAeasy minikit. Turbo DNase kit (Ambion) was used to remove contaminating DNA. Bacterial content of patient samples was enriched using MICROBEnrich kit (Ambion), which depletes RNA of eukaryotic mRNA and

rRNA. Library preparation and sequencing was performed by University of Michigan sequencing core. ScriptSeq Complete Kit (Bacteria) library kit was used to both deplete samples of bacterial rRNA and to construct stranded cDNA libraries from the rRNA-depleted RNA (*Table 3*, *Table 4*). While the original in vitro samples submitted for sequencing were not treated with MICROBEnrich kit, we have since performed extensive testing with two different clinical UTI strains (HM86 and HM56) to show that treatment with the kit does not affect the measured gene expression (*Figure 1—figure supplement 5*, *Supplementary file 1*). All samples were sequenced using Illumina HiSeq2500 (single end, 50 bp read length).

## RT-PCR validation of MICROBEnrich-treated samples

Clinical strains HM56 and HM86 were cultured overnight in LB broth at 37°C. The next morning, the culture was spun down, and the pellet washed once with PBS. Pooled filter-sterilized human urine was then inoculated with the washed bacteria at a ratio of 1:100 and incubated shaking at 37°C for five hours. Cultures were then treated with bacterial RNAprotect (Qiagen), pellets collected and stored at −80°C. The bacterial pellets were treated with both lysozyme and proteinase K, and then total RNA was extracted using the RNAeasy kit (Qiagen). Genomic DNA was removed using the Turbo DNA free kit (ThermoFisher). The extracted RNA was then halved. One half was treated using the MICROBEnrich kit (ThermoFisher), which should only remove eukaryotic mRNA and eukaryotic rRNA. The second half of the RNA remained untreated. Both the MICROBEnrich and untreated samples were reverse-transcribed into cDNA using the iScript cDNA synthesis kit (Biorad), with 1 µg RNA as template. Real-Time Quantitative Reverse Transcription PCR (qRT-PCR) was performed in a Quantstudio 3 PCR system (Applied Biosystem) in technical triplicate, using SYBR green (ThermoFisher). Samples were normalized to *gapA* transcript levels, by subtracting the Ct values of *gapA* from the Ct values of monitored genes. This value is reported as ΔCt.

## Characterization of virulence factors' gene expression

We compiled a literature search-based list of virulence factors belonging to different functional groups. Sequences for each virulence factor gene were extracted from reference UPEC genomes (either CFT073 or UTI89). Presence or absence of each virulence factor within clinical genomes was determined using BLAST (with percent identity $\geq$80% and percent coverage $\geq$90%, e-value $\leq$10$^{-6}$). Hierarchical clustering of strains based on presence or absence of virulence factors was performed using Python's scipy.cluster.hierarchy.linkage function with default parameters. Heatmaps of virulence factors' gene expression in urine and in patients show normalized transcripts per million (TPMs) (same as for correlation analysis and PCA, see below).

## RNAseq data processing

A custom bioinformatics pipeline was used for the analysis (*Sintsova, 2019*; copy archived at https://github.com/elifesciences-publications/rnaseq_analysis). Raw fastq files were processed with Trimmomatic (*Bolger et al., 2014*) to remove adapter sequences and analyzed with FastQC to assess sequencing quality. Mapping was done with bowtie2 aligner (*Langmead and Salzberg, 2012*) using default parameters. Alignment details can be found in *Table 3* and *Table 4*. Read counts were calculated using HTseq htseq-count (union mode) (*Anders et al., 2015*).

## Quality control

Since some of our clinical samples yielded lower numbers of bacterial reads than desired (*Table 3*), we performed a comprehensive quality assurance to determine if the sequencing depth of our clinical samples was sufficient for our analysis (see Saturation curves and Gene expression ranges analysis below, *Figure 2—figure supplement 1*, *Figure 2—figure supplement 2*). Overall, all patient samples except for HM66 passed quality control (see gene expression ranges analysis, *Figure 2—figure supplement 2*). While we elected to keep all of the strains in our subsequent analysis, this observation explains why the patient HM66 sample appears as an outlier in *Figure 2*.

## Saturation curves

We created saturation curves for each of our sequencing files to assess whether we have sufficient sequencing depth for downstream analysis. Each sequencing file was subsampled to various degrees

and number of genes detected in those subsamples (y-axis) was graphed against number of reads in the subsample (x-axis). As expected, all of the in vitro samples reached saturation (*Figure 2—figure supplement 1*, blue lines). Unfortunately, 6 out of our 14 samples did not reach saturation, which warranted us to investigate further (see Gene expression ranges analysis) *Figure 2—figure supplement 1*, red lines). Additionally, dropping the six samples that did not reach saturation from our analysis did not affect any of the results.

## Core genome identification

Core genome for 14 clinical isolates and MG1655 was determined using get_homologues (*Contreras-Moreira and Vinuesa, 2013*). We explored multiple parameter values for our analysis and their effect on final core genome, in the end we set the cut off of 90% of sequence identity and 50% sequence coverage (similar results were obtained when using different cutoffs). The intersection of three algorithms employed by get_homologues contained 2653 gene clusters.

## Gene expression ranges analysis

Due to low sequencing depth of 6 of our isolates, we were worried we would not be able to detect genes expressed at low levels in those samples. To evaluate whether we were losing information about low-level expression, we compared a number of genes in the core genome that were expressed at different levels (1000 TPMS, 100 TPMS, 10 TPMS and 1 TPM) between clinical samples that reached saturation (*Figure 2—figure supplement 2A*) and those that did not (*Figure 2—figure supplement 2B*). Only one of the clinical samples (HM66) seemed to lack genes expressed in the range of 1–10 TPMs. Thus, we conclude that all but one sample (HM66) had sufficient coverage for downstream analysis.

## Pearson correlation coefficient calculation and PCA analysis

For PCA and correlation analysis, transcript per million (TPM) was calculated for each gene, TPM distribution was then normalized using inverse rank transformation. Pearson correlation and PCA was performed using python Python sklearn library. Jupyter notebooks used to generate the figures are available at https://github.com/ASintsova/HUTI-RNAseq.

## Differential expression analysis

Differential expression analysis was performed using DESeq2 R package (*Love et al., 2014*). Genes with log2 fold change of greater than two or less than −2 and adjusted *p* value (Benjamini-Hochberg adjustment) of less than 0.05 were considered to be differentially expressed. DESeq2 normalized counts were used to generate *Figure 3* and *Figure 6*. Pathway analysis was performed using R package topGO (*Alexa and Rahnenfuhrer, 2018*).

## RNA sequencing of HM43 from the mouse model of UTI

Forty CBA/J mice were infected using the ascending model of UTI as previously described (*Hagberg et al., 1983*). Briefly, 40 six-week-old female mice were transurethrally inoculated with $10^8$ CFU of UPEC isolate HM43. 48 hr post infection urine was collected from each mouse directly into bacterial RNAprotect (Qiagen). All collected urine was pooled together and pelleted, and immediately placed in the −80°C freezer. This collection was repeated every 45 minutes five more times, resulting in six collected pellets consisting of bacterial and eukaryotic cells.

For in vitro controls, UPEC strain HM43 was cultured overnight in LB. The next morning, the culture was spun down, and the pellet washed twice with PBS. LB or pooled human urine was then inoculated with the washed bacteria at a ratio of 1:100 and incubated with shaking at 37°C for 3 hr. Cultures were then treated with bacterial RNAprotect (Qiagen), pellets collected and stored at −80°C.

All the pellets were treated with both lysozyme and proteinase K, and then total RNA was extracted using RNAeasy kit (Qiagen). Genomic DNA was removed using the Turbo DNA free kit (ThermoFisher). Eukaryotic mRNA was depleted using dynabeads covalently linked with oligo dT (ThermoFisher). The in vitro samples underwent the same treatment with dynabeads to reduce any potential biases this procedure might introduce to the downstream sequencing. The supernatant was collected from this treatment, and the RNA was concentrated and re-purified using RNA Clean

and Concentrator kit (Zymo). Library preparation and sequencing was performed by University of Michigan sequencing core. The ScriptSeq Complete Gold Kit (Epidemiology) library kit was used to both deplete samples of bacterial and eukaryotic rRNA and to construct stranded cDNA libraries from the rRNA-depleted RNA. These were sequenced using Illumina HiSeq2500 (single end, 50 bp read length). RNAseq analysis was performed as described above, alignment statistics are shown in *Table 8* and *Table 9*.

Analysis of RNAseq data from *Subashchandrabose et al. (2014)*. Sample collection and RNA isolation is described in *Subashchandrabose et al. (2014)*. Briefly, RNA samples were treated with proteinase K and total RNA was isolated using Qiagen RNAeasy minikit. Turbo DNase kit (Ambion) was used to remove contaminating DNA. Bacterial content of patient samples was enriched using MICROBenrich kit (Ambion). The depleted RNA was used to generate sequencing libraries using the Ovation Prokaryotic RNA-Seq system (NuGen) and the Encore next-generation sequencing library system (NuGen). The libraries were sequenced using an Illumina HiSeq2000 (paired-end, 100 bp) by the Genome Resource Center at the Institute for Genome Sciences, University of Maryland, Baltimore, MD. RNAseq analysis was performed as described above, alignment statistics are shown in *Table 6* and *Table 7*.

## Estimation of HM43 doubling time

For both LB and urine OD curves were performed using Bioscreen-C Automated Growth Curve Analysis System (Growth Curves USA) eight separate times. For each time point, the mean values of the eight replicates were used for doubling time estimation. The equation bellow was used to estimate doubling time during logarithmic growth in LB or urine, where DT is doubling time, C2 is final OD, C1 is initial OD, and $t$ is time elapsed between when C2 and C1 were taken.

$$DT = \frac{t * log2}{log(C2) - log(C1)}$$

DT was calculated for every two measurements taken between 30 and 180 min and mean of these values is reported.

## Regulon analysis

Regulon gene sets were extracted from RegulonDB 9.4 (*Gama-Castro et al., 2016*) using custom Python scripts (available https://github.com/ASintsova/HUTI-RNAseq). Gene set enrichment analysis was performed using Python GSEAPY library.

## Data access

Jupyter notebooks as well as all the data used to generate the figures in this paper are available on github: https://github.com/ASintsova/HUTI-RNAseq.

---

## Additional information

### Funding

| Funder | Grant reference number | Author |
|---|---|---|
| National Institute for Health Research | R01 DK094777 | Anna Sintsova<br>Arwen E Frick-Cheng<br>Sara Smith<br>Ali Pirani<br>Sargurunathan Subashchandrabose<br>Evan S Snitkin<br>Harry Mobley |
| American Urological Association Foundation | Research Scholar Fellow | Anna Sintsova |

The funders had no role in study design, data collection and interpretation, or the decision to submit the work for publication.

## Author contributions

Anna Sintsova, Conceptualization, Data curation, Formal analysis, Validation, Investigation, Visualization, Writing—original draft; Arwen E Frick-Cheng, Validation, Investigation, Visualization, Writing—review and editing; Sara Smith, Validation, Investigation; Ali Pirani, Data curation, Formal analysis, Validation; Sargurunathan Subashchandrabose, Investigation, Methodology, Writing—review and editing; Evan S Snitkin, Conceptualization, Formal analysis, Supervision, Methodology, Writing—review and editing; Harry Mobley, Conceptualization, Formal analysis, Funding acquisition, Investigation, Methodology, Project administration, Writing—review and editing

## Author ORCIDs

Anna Sintsova (iD) https://orcid.org/0000-0003-4075-6366
Arwen E Frick-Cheng (iD) https://orcid.org/0000-0002-7202-4701
Harry Mobley (iD) https://orcid.org/0000-0001-9195-7665

## Ethics

Human subjects: All procedures involving human samples were performed in accordance with the protocol (HUM00029910) approved by the Institutional Review Board at the University of Michigan. This protocol is compliant with the guidelines established by the National Institutes of Health for research using samples derived from human subjects.

Animal experimentation: Mouse infection experiments were conducted according to the protocol PRO00007111 approved by the University Committee on Use and Care of Animals at the University of Michigan. This protocol is in complete compliance with the guidelines for humane use and care of laboratory animals established by the National Institutes of Health.

## Decision letter and Author response

Decision letter https://doi.org/10.7554/eLife.49748.039
Author response https://doi.org/10.7554/eLife.49748.040

# Additional files

## Supplementary files

• Supplementary file 1. Primers used for qPCR experiments.
DOI: https://doi.org/10.7554/eLife.49748.032

• Transparent reporting form DOI: https://doi.org/10.7554/eLife.49748.033

## Data availability

Sequencing data have been deposited in GEO under accession codes GSE128997.

The following dataset was generated:

| Author(s) | Year | Dataset title | Dataset URL | Database and Identifier |
|---|---|---|---|---|
| Sintsova A, Frick-Cheng A, Smith S, Pirani A, Snitkin E, Mobley H | 2019 | Genetically diverse uropathogenic *Escherichia coli* adopt a common transcriptional program in patients with urinary tract infections | https://www.ncbi.nlm.nih.gov/geo/query/acc.cgi?acc=GSE128997 | NCBI Gene Expression Omnibus, GSE128997 |

The following previously published dataset was used:

| Author(s) | Year | Dataset title | Dataset URL | Database and Identifier |
|---|---|---|---|---|
| Subashchandrabose S, Hazen TH, Brumbaugh AR, Himpsl SD, Smith SN, Ernst RD, Rasko DA, Mobley HLT | 2014 | *Escherichia coli* HM26 Transcriptome or Gene expression | https://www.ncbi.nlm.nih.gov/sra?term=SRP041701 | NCBI Sequence Read Archive, SRP041701 |

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
