## [Decision Letter]

[Editors’ note: a previous version of this study was rejected after peer review, but the authors submitted for reconsideration. The first decision letter after peer review is shown below.]

Thank you for submitting your work entitled "Genetically diverse *Escherichia coli* adopt a common transcriptional program in patients with urinary tract infections" for consideration by *eLife*. Your article has been reviewed by three peer reviewers, and the evaluation has been overseen by a Reviewing Editor and a Senior Editor. The following individual involved in review of your submission has agreed to reveal their identity: Alexander Westermann (Reviewer #2).

Our decision has been reached after consultation between the reviewers. Based on these discussions and the individual reviews below, we regret to inform you that your work will not be considered further for publication in *eLife*. All reviewers agreed on the importance of your work on characterising the transcriptional program of strains directly from patients. However, they raised serious technical questions with respect to the different processing of the RNA samples from patients, mice and in vitro, and potential confounding factors that cannot be eliminated without extensive new experiments. If you wished to address all the reviewers' concerns, we would be willing to assess the suitability for publication of an extensively revised version of this manuscript.

Reviewer #1:

This manuscript continues a large body of work by Mobley and co-workers to define a UPEC virulence genotype during human UTI. The authors use RNAseq to characterise a transcriptional program shared by a genetically diverse group of 14 UPEC strains harvested directly from the urine of infected patients. This revealed a signature defining upregulation of genes involved in translation, and demonstrating that UPEC grow rapidly during human UTI.

A major strength of this work is the transcriptome analysis of UPEC growth during human UTI, and comparison of this to growth in vitro and growth in the mouse model of UTI. The conserved transcriptional signature of the core genome during human UTI led to the important discovery that reprogramming occurs, and results in the allocation of cellular resources to support rapid growth during human UTI. It was comforting to see a conserved profile of transcription in humans and mice, an important finding that will be of great value to the field. In a broader context, this work provides new insight into UPEC adaptation to the human urinary tract. The transcriptional program that defines rapid growth during human UTI provides a framework to design novel therapeutics that block this phenotype in UPEC, an urgent need in the context of rapidly increasing antibiotic resistance.

Overall, I have no substantive concerns regarding the data or major conclusions.

Reviewer #2:

In the present study, Sintsova et al. profile expression of the core genome of 14 UPEC isolates right upon their collection from UTI patients and compare it with the corresponding bacterial transcriptome patterns when grown in vitro. In doing so, the authors identify a conserved expression pattern shared among all isolates, which is associated with an increased expression of mRNAs encoding ribosomal proteins and a reduced expression of metabolic genes in the in vivo isolates compared to the in vitro cultures. Based on this finding, the authors conclude that in vivo UPEC reallocates its resources to increase proliferation at the expense of metabolic activity, and consequently, that bacterial growth may be enhanced in the patient's bladder as compared to in vitro conditions. This would imply that one or several elusive host factors (present in the bladder but absent from the sterile-filtered urine) would enhance UPEC growth.

My major concern is that the study lacks enough data to support this hypothesis. The growth rate is deduced from the number of sequencing reads that map to genes for ribosomal proteins. This measure happens to fit with the growth behavior in vitro, when the authors compare UPEC replication in LB and sterile urine. However, this does not necessarily mean that the same applies in vivo. Given that this is their major finding, the authors may want to further support this speculation by in vivo data, e.g. by determining CFU counts in their mouse model over time of infection. Also, what would speak against their hypothesis is that genes for other cellular functions required for growth (such as DNA replication, cell division, etc.) appear not to be differentially expressed in vivo vs. in vitro.

Also with respect to the generation and analysis of RNAseq data, I have some comments:

– It remains unclear if the authors sequenced bacterial total RNA or rRNA-depleted RNA. The MICROBEnrich kit used, depletes polyadenylated transcripts (eukaryotic mRNAs and certain lincRNAs) as well as eukaryotic rRNAs, but does not efficiently deplete bacterial rRNA. Also the Illumina ScriptSeq v2 kit per se, does not deplete ribosomal transcripts. Therefore, it appears unclear why the authors state that "… rRNA-depleted stranded cDNA libraries…" were constructed. By the way, if indeed not actively depleted, reads mapping to bacterial rRNAs should also be increased in the in vivo samples (as are reads mapping to ribosomal proteins). In general, plots or tables that inform about RNA class distributions (% reads mapping to mRNAs, rRNAs, tRNAs, etc.) in the individual samples would be helpful.

– Why didn't the authors include published datasets in their analysis? This would seem particularly obvious for the data derived from their own previous (pilot) study (Subashchandrabose et al., 2014) that was based on 5 samples taken together with the 14 samples analyzed here.

Reviewer #3:

The manuscript "Genetically diverse uropathogenic *Escherichia coli* adopt a common transcriptional program in patients with urinary tract infections", by Sintsova et al., presents an RNAseq-driven analysis of UPEC gene expression from 14 UTI patients. The main experiment is to compare the expression of the bacteria directly isolated from the patients to the expression of the bacteria after growth in filter-sterilized, pooled human urine in vitro. The primary result is that bacteria isolated from the urine of patients have high expression of genes encoding proteins involved in DNA and protein synthesis: ribosomal proteins, rRNA and tRNA modification proteins, purine and pyrimidine metabolism.

The primary result was examined with a few analyses on the patient vs. in vitro urine expression. The analysis was split early between virulence genes/accessory genes and "core" genes that are present in all the strains. The analysis focused relatively quickly on the core genes, which included all the DNA and protein synthesis genes noted above. Further analysis of the core genes resulted in a few other general features that differentiated patient (infection) from in vitro growth: downregulation of amino acid biosynthesis, downregulation of sugar metabolism, downregulation of most sugar transporters, and upregulation of amino acid transporters.

An important set of validation experiments was then done with one strain, HM43, which was grown in LB (considered a "rich" lab media), a new batch of filter-sterilized urine, and multiple urine samples from mice that had been infected in their bladders with HM43. Using% of reads mapping to genes encoding ribosomal proteins and to genes encoding catabolic enzymes showed that the in vivo mouse infection samples again had a high proportion of ribosomal reads and low catabolic reads, more similar to the human patient expression data and not matched by the LB-grown bacteria. This last comparison was the main test the effect of a faster growth rate per se.

The authors then conclude that there is an infection-specific transcriptional program which is dedicated to high growth rate in urine. They further noted that the downstream regulated genes for 22 transcriptional factors were differentially expressed between patient infections and in vitro urine growth, and speculate that some environmental cues may be sensed and thereby lead to the observed high expression of DNA and protein synthesis genes.

I have one primary technical question about the experimental design and a question about the context within the UTI field. First, the authors have had substantial experience with doing RNAseq from patient urines, and take care to attempt to minimize the time between urine sample collection and RNAProtect addition (to be applauded). Also, all samples according to the Materials and methods are also stored in RNAProtect. One remaining issue is that it states that the "bacterial content of patient samples was enriched using MICROBEnrich kit". This raises a potential confounding variable that seems reasonable for explaining a large scale consistent different between patient samples and in vitro urine samples. For the HM43 mouse experiment, it is stated that "eukaryotic mRNA was depleted using dynabeads covalently linked with oligodT". This would seem to leave the eukaryotic ribosomal RNA still in the sample, but sequencing and mapping statistics for this experiment (similar to Table 1 for the human samples) are not included to check on this. Therefore, I am wondering whether the in vivo mouse samples were also treated differently than the in vitro LB and urine samples. Regardless, the authors should clarify the methods particularly for whether the MICROBEnrich was used only on the patient samples and explicitly not on the in vitro urine samples, and also they should similarly provide a bit more detail on the processing of the mouse samples.

In addition, the primary result from this paper has largely been described before. Bielecki et al. (2014) performed RNAseq on 21 strains from human UTI patients, 4 of which were subsequently grown in LB for to get an in vitro RNAseq data set; also of note these authors used MICROBEnrich for host RNA removal on the human patient samples, and they did rRNA depletion with a MICROBExpress kit for the in vitro samples. I suggest that the authors more explicitly acknowledge this previous work and the general observations that have already been made, which will help them to either better differentiate their current study or strengthen the overall result shared by the two papers by providing additional validating data.

[Editors’ note: what now follows is the decision letter after the authors submitted for further consideration.]

Thank you for submitting your article "Genetically diverse uropathogenic *Escherichia coli* adopt a common transcriptional program in patients with UTIs" for consideration by *eLife*. Your article has been reviewed by three peer reviewers, and the evaluation has been overseen by a Reviewing Editor and Neil Ferguson as the Senior Editor. The following individual involved in review of your submission has agreed to reveal their identity: Alexander Westermann (Reviewer #2).

The reviewers have discussed the reviews with one another and the Reviewing Editor has drafted this decision to help you prepare a revised submission.

The work presented here is a revised version of a previous submission. The original paper was deemed very interesting by reviewers, who however had raised several key issues, notably potential confounding of results due to differences in in vitro vs. in vivo RNA isolation/processing in the lab. The reviewers now agree that you have provided substantial new data to address these concerns and would be willing to consider this manuscript positively for publication. However, there are a list of points that should be addressed before this could be considered.

Essential revisions:

1) Could you explain how you selected the extra dataset provided in Figure 1—figure supplement 5 that suggests that MICROBEnrich has little effect on bacterial gene expression? Importantly, were these some of the genes that showed differential expression between the in vivo and in vitro samples? It is essential to include, if not already tested, at least some of the r-protein-encoding mRNAs in this analysis given their importance in the study.

2) Table 3: Only a small percentage of the reads from the in vivo samples map to the UPEC genome. Could you explain what the remainder of the reads might be derived from? Would they map to the host genome (thus indicating that bacterial enrichment is rather low-efficient), or is there any evidence that some of those non-UPEC reads might be derived from other causative agents of UTI?

---

## [Author Response]

[Editors’ note: the author responses to the first round of peer review follow.]

Reviewer #2:[…] My major concern is that the study lacks enough data to support this hypothesis. The growth rate is deduced from the number of sequencing reads that map to genes for ribosomal proteins. This measure happens to fit with the growth behavior in vitro, when the authors compare UPEC replication in LB and sterile urine. However, this does not necessarily mean that the same applies in vivo. Given that this is their major finding, the authors may want to further support this speculation by in vivo data, e.g. by determining CFU counts in their mouse model over time of infection. Also, what would speak against their hypothesis is that genes for other cellular functions required for growth (such as DNA replication, cell division, etc.) appear not to be differentially expressed in vivo vs. in vitro.

We appreciate the reviewers concerns and apologize for lack of clarity in our discussion of data on growth rate:

1) As mentioned by the reviewer, levels of ribosomes/rRNA are known to be closely correlated with growth rate under many different experimental conditions (Basan, 2018; Basan et al., 2015; Molenaar et al., 2009; Scott et al., 2010; Scott and Hwa, 2011; You et al., 2013), and there is no reason to believe that this would not also hold true in an in vivosetting. Additionally, while, in the past, we have measured CFUs during infection, bacterial load cannot be used as a proxy for growth rate, as it will be affected by multiple factors besides bacterial growth rate (such as bacterial clearance by the immune response to the infection).

2) We apologize that this was not clear from our text, but in fact we do show that other genes required for growth are upregulated during infection (Figure 3B, subsection “UPEC show increased expression of replication and translation machinery during UTI”, first paragraph). In addition to genes mentioned in Figure 3, there are other genes required for growth that are listed in Figure 3—source data 1, such as *fis, dnaG* (DNA replication), *dbpA* (ribosome biogenesis)*, mrcA,* and *mrdA* (peptidoglycan biosynthesis). Additionally, while there are sure to be other genes/proteins that are required for growth that were not significantly upregulated during infection, they might be regulated at a translational/post-translational level, and thus would not be changed in their expression.

3) Our conclusions about rapid growth rate are not only based on the gene expression profile observed in this study, but are also supported by previous studies that attempt to directly measure UPEC growth rate in patients with UTIs (Burnham et al., 2018; Forsyth et al., 2018). We have tried to make this clear throughout the manuscript (specifically in the last paragraph of the Introduction, the first paragraph of the subsection “UPEC show increased expression of replication and translation machinery during UTI” and in the third paragraph of the Discussion). In this study we show how this rapid growth rate can result from repartitioning of the bacterial transcriptome that is observed both during mouse and human UTI.

Also with respect to the generation and analysis of RNAseq data, I have some comments:– It remains unclear if the authors sequenced bacterial total RNA or rRNA-depleted RNA. The MICROBEnrich kit used, depletes polyadenylated transcripts (eukaryotic mRNAs and certain lincRNAs) as well as eukaryotic rRNAs, but does not efficiently deplete bacterial rRNA. Also the Illumina ScriptSeq v2 kit per se, does not deplete ribosomal transcripts. Therefore, it appears unclear why the authors state that "… rRNA-depleted stranded cDNA libraries…" were constructed. By the way, if indeed not actively depleted, reads mapping to bacterial rRNAs should also be increased in the in vivo samples (as are reads mapping to ribosomal proteins). In general, plots or tables that inform about RNA class distributions (% reads mapping to mRNAs, rRNAs, tRNAs, etc.) in the individual samples would be helpful.

We apologize for the lack of clarity and omission of RNA class distribution data. We have revised our Materials and methods section (subsection “RNA isolation and sequencing”) and added Tables 3 and 4 to address this point. These tables show clear depletion of bacterial rRNA from our samples. Specifically, both eukaryotic and prokaryotic rRNA was depleted from the samples prior to sequencing. Bacterial content of patient samples was enriched using MICROBEnrich kit (Ambion), which depletes the sample of eukaryotic mRNA and rRNA. Library preparation and sequencing was performed by University of Michigan sequencing core. ScriptSeq Complete Kit (Bacteria) library kit (https://www.illumina.com/products/scriptseq-bacteria.html) was used to both deplete samples of bacterial rRNA and to construct stranded cDNA libraries from the rRNA-depleted RNA.

– Why didn't the authors include published datasets in their analysis? This would seem particularly obvious for the data derived from their own previous (pilot) study (Subashchandrabose et al., 2014) that was based on 5 samples taken together with the 14 samples analyzed here.

The samples from our pilot study were not analyzed alongside this study because the library preparation, sequencing and facility used in the pilot study were different. However, we appreciate this comment, which has also been brought up by reviewer #3. We have now analyzed data from Subashchandrabose (2014) and show that it is consistent with our current study, i.e., increase intranscripts dedicated to r-protein production in the patient samples compared to LB or urine. These data have now been added as Figure 4—figure supplement 1 and are discussed in the last paragraph of the subsection “A shift in metabolic gene expression during UTI to optimize growth potential”, and in the subsection “Analysis of RNAseq data from Subashchandrabose et al.” One of the patient samples (HM26) had very few reads that mapped to the bacterial genome (Table 6, 7), which potentially explains the fact that this was the only patient sample that contained very few r-protein reads.

Reviewer #3:[…] I have one primary technical question about the experimental design and a question about the context within the UTI field. First, the authors have had substantial experience with doing RNAseq from patient urines, and take care to attempt to minimize the time between urine sample collection and RNAProtect addition (to be applauded). Also, all samples according to the Materials and methods are also stored in RNAProtect. One remaining issue is that it states that the "bacterial content of patient samples was enriched using MICROBEnrich kit". This raises a potential confounding variable that seems reasonable for explaining a large scale consistent different between patient samples and in vitro urine samples. For the HM43 mouse experiment, it is stated that "eukaryotic mRNA was depleted using dynabeads covalently linked with oligodT". This would seem to leave the eukaryotic ribosomal RNA still in the sample, but sequencing and mapping statistics for this experiment (similar to Table 1 for the human samples) are not included to check on this. Therefore, I am wondering whether the in vivo mouse samples were also treated differently than the in vitro LB and urine samples. Regardless, the authors should clarify the methods particularly for whether the MICROBEnrich was used only on the patient samples and explicitly not on the in vitro urine samples, and also they should similarly provide a bit more detail on the processing of the mouse samples.

1) We sincerely apologize for the lack of clarity in our Materials and methods, as we did not state clearly whether eukaryotic/prokaryotic rRNA was depleted prior to RNA sequencing. We have expanded the Materials and methods section to clearly state that indeed both eukaryotic and prokaryotic rRNA was depleted from all of our samples sequenced in this study (subsection “RNA isolation and sequencing”), and provide additional tables showing numbers of reads that mapped to different types of RNA for each sample, for both clinical, in vitro and mouse samples (Tables 3, 4, 8, and 9).

2) Additionally, we also failed to clearly explain that for the mouse study described in Figure 5 all samples (in vitroLB and urine cultures, as well as RNA isolated from mice with UTIs) were subjected to the same protocol (i.e., treatment with dynabeads covalently linked to oligo dT) to avoid any potential biases that could result from differential RNA processing. This clarification has now been added in the last paragraph of the subsection “RNA sequencing of HM43 from mouse model of UTI”.

3) The reviewer brought up a very important point suggesting that treatment of patient samples with MICROBEnrich could result in differential gene expression program we observed in the study. We want to point out that if treatment with MICROBEnrich affected the measurement of gene expression or RNA pool composition, we would expect to see large-scale global difference in gene expression, when in fact less than 500 genes were differentially expressed between the two conditions and were enriched in specific biological processes. However, we have taken this concern very seriously and provide five separate lines of evidence that the gene expression pattern observed in patient population is not due to an RNA processing step. These are summarized above, and are further elucidated below:

- We have performed an extensive RT-PCR study to show that treatment with MICROBEnrich does not change the levels of gene expression in multiple UPEC strains cultured in urine. Specifically, we isolated RNA from urine cultures of HM86 and HM56, split each of the samples in half, where one half went through MICROBEnrich treatment, and the other was left untreated. We then measured gene expression for a panel of 10 different genes for both treated and untreated samples by RT-PCR and found no differences in gene expression between any of them. This was done in three biological replicates for HM86, and two biological replicates for HM56. This provides strong evidence that treatment with MICROBEnrich alone does not affect measurement of gene expression. These experiments are now explained in the Materials and methods subsection “RT-PCR validation of MICROBEnrich-treated samples” and presented in Figure 1—figure supplement 5, Table 10, and Supplementary file 1.

- As discussed above, for the mouse UTI experiments, in vitroand in vivo RNA samples were treated exactly the same (yet differently from the original urine and patient samples). Nevertheless, the gene expression during mouse UTI highly resembles that of human UTI. This excludes the possibility of our results being attributable to MICROBEnrich.

- In the revised Figure 1, we take a closer look at expression of genes that we expect to be similarly highly expressed in urine and patients (i.e., iron acquisition genes). As expected, we see a similar pattern of expression for a number of different virulence factors between in vitroand patient samples, which argues against a MICROBEnrich treatment having an effect on measured gene expression. This data is now presented in Figure 1B and Figure 1—figure supplement 3 and is described in the Results subsection “Virulence factor expression is not specific to infection”.

- One of the main conclusions of the paper (i.e., conserved gene expression program during UTI) does not rely on the comparison between in vitroand patient samples, and therefore the difference in treatment between these two sample groups would have minimal impact on our stated results. In fact, we first observed the conserved gene expression program when we compared gene expression of core genome and accessory genome between patients and observed that the core gene expression is more conserved. We apologize for omitting these data from the original manuscript, it is now included in Figure 2A and 2B and is discussed in the first paragraph of the subsection “The UPEC core genome exhibits a common gene expression program during clinical Infection”.

- Finally, we believe our original manuscript did not stress enough the importance of previous work on UPEC growth rate during infection. Specifically, Bielecki et al.(2014) have also observed a gene expression pattern consistent with fast growth in patients with UTIs. In addition, two other studies attempt to directly measure UPEC growth rate in patients with UTI and find it to be consistently fast [(Burnham et al., 2018; Forsyth et al., 2018), the latter from our lab]. The fact that multiple studies across different patient cohorts came to similar conclusions argues against our study being affected by differential MICROBEnrich treatment. We have revised the manuscript to highlight the previous work done by us and others throughout the Results and Discussion. Specifically, the following was modified: Introduction, last paragraph; subsection “UPEC show increased expression of replication and translation machinery during UTI”, first paragraph; Discussion, third and sixth paragraphs.

In addition, the primary result from this paper has largely been described before. Bielecki et al. (2014) performed RNAseq on 21 strains from human UTI patients, 4 of which were subsequently grown in LB for to get an in vitro RNAseq data set; also of note these authors used MICROBEnrich for host RNA removal on the human patient samples, and they did rRNA depletion with a MICROBExpress kit for the in vitro samples. I suggest that the authors more explicitly acknowledge this previous work and the general observations that have already been made, which will help them to either better differentiate their current study or strengthen the overall result shared by the two papers by providing additional validating data.

As discussed above, we are aware of the importance of the study described here, and it was not our intention to discount it in any way. In fact, we think that the fact that the two studies are in close agreement, despite studying different patient cohorts, is remarkable and significantly strengthens the conclusions of both studies. We have revised the text of the manuscript to more explicitly acknowledge this work, specifically the following was modified: Introduction, last paragraph; subsection “UPEC show increased expression of replication and translation machinery during UTI”, first paragraph; Discussion, third and sixth paragraphs. We also want to emphasize that the purpose of our study was also to build on these findings and provide a mechanistic insight into how this growth rate could potentially be achieved. Specifically, we show how differential transcriptome partitioning can facilitate the rapid growth. Moreover, we recapitulate our observations in the mouse model of UTI and characterize *E. coli* regulatory network that is behind the transcriptome partitioning observed during human UTI.

[Editors' note: the author responses to the re-review follow.]Essential revisions:1) Could you explain how you selected the extra dataset provided in Figure 1—figure supplement 5 that suggests that MICROBEnrich has little effect on bacterial gene expression? Importantly, were these some of the genes that showed differential expression between the in vivo and in vitro samples? It is essential to include, if not already tested, at least some of the r-protein-encoding mRNAs in this analysis given their importance in the study.

For the original panel, we selected genes that have previously been shown to be important for UPEC infections. However, they were not differentially expressed between in vivoand in vitroconditions. We have now extended this panel by including data for five more genes, 2 of which (*nanM* and *malK*) were downregulated under in vivo conditions, and 3 of which (ribosomal protein genes *rpoA, rplA, rpsA*) were upregulated in vivo. As is shown in Supplementary file 1, MICROBEnrich treatment did not affect measurements of gene expression for any of the genes in the panel.

2) Table 3: Only a small percentage of the reads from the in vivo samples map to the UPEC genome. Could you explain what the remainder of the reads might be derived from? Would they map to the host genome (thus indicating that bacterial enrichment is rather low-efficient), or is there any evidence that some of those non-UPEC reads might be derived from other causative agents of UTI?

The majority of the reads from those samples do indeed map to the host genome. Moreover, none of the patients in this study had polymicrobial infections. While the enrichment was not perfect, in our hands, it does significantly increase the amount of bacterial RNA in our samples.